# Population-scale peach genome analyses unravel selection patterns and biochemical basis underlying fruit flavor

Yang Yu [1,4], Jiantao Guan[1,4], Yaoguang Xu[1,4], Fei Ren[2,4], Zhengquan Zhang[1], Juan Yan[3], Jun Fu [1], Jiying Guo[2], Zhijun Shen[3], Jianbo Zhao[2], Quan Jiang[2✉], Jianhua Wei[1✉] & Hua Xie [1✉]

A narrow genetic basis in modern cultivars and strong linkage disequilibrium in peach (*Prunus persica*) has restricted resolution power for association studies in this model fruit species, thereby limiting our understanding of economically important quality traits including fruit flavor. Here, we present a high-quality genome assembly for a Chinese landrace, Longhua Shui Mi (LHSM), a representative of the Chinese Cling peaches that have been central in global peach genetic improvement. We also map the resequencing data for 564 peach accessions to this LHSM assembly at an average depth of 26.34× per accession. Population genomic analyses reveal a fascinating history of convergent selection for sweetness yet divergent selection for acidity in eastern vs. western modern cultivars. Molecular-genetics and biochemical analyses establish that PpALMT1 (aluminum-activated malate transporter 1) contributes to their difference of malate content and that increases fructose content accounts for the increased sweetness of modern peach fruits, as regulated by PpERDL16 (early response to dehydration 6-like 16). Our study illustrates the strong utility of the genomics resources for both basic and applied efforts to understand and exploit the genetic basis of fruit quality in peach.

[1] Beijing Agro-Biotechnology Research Center, Academy of Agriculture and Forestry Sciences/Beijing Key Laboratory of Agricultural Genetic Resources and Biotechnology, Beijing, China. [2] Beijing Academy of Forestry and Pomology Sciences, Beijing Academy of Agriculture and Forestry Sciences, Beijing, China. [3] Institute of Pomology, Jiangsu Academy of Agricultural Sciences/Jiangsu Key Laboratory for Horticultural Crop Genetic Improvement, Nanjing, China. [4] These authors contributed equally: Yang Yu, Jiantao Guan, Yaoguang Xu, Fei Ren. ✉email: quanj@vip.sina.com; weijianhua@baafs.net.cn; xiehua@baafs.net.cn

Fruits are an indispensable component of healthy human diets, providing vitamins, minerals, dietary fibers, antioxidants, and calories[1]. Sweetness and acidity are two of the important flavor determinants which influence consumer preference and acceptability[2]. Current genome researches have strengthened the genetic basis underlying these two internal quality properties for fruit flavor improvement in many fruit crops[3–7].

Domesticated peach (*Prunus persica* (L.) Batsch), a model for genetics and genomics of the genus *Prunus* and other related Rosaceae perennial fruit crops[8] especially in study on the formation mechanism of fruit quality[9], originated in China over two million years ago (MYA)[10,11] and had undergone thousands of years' cultivation and improvement, particularly for fruit quality in China[12,13]. Chinese peach germplasm has been foundational in the development of virtually all modern peach cultivars[12]. Two phases of peach dispersal from China have together profoundly impacted the genetic diversity of modern cultivars worldwide: an initial dispersal of primitive peach landraces (presumably) from northwestern China (dating from the final centuries BC) and the later dispersal of landraces with excellent fruit quality (particularly low-acid and sweet peaches) from eastern China (dating from mid-19th century) to locations around the world[12,14–17]. It is notable that current preferences for peach flavors differ substantially around the world, forming two typical flavor types: sweet, low-acid vs. sweet, acid taste, respectively favored by eastern and western consumers[18,19]. However, molecular mechanisms that explain how past genetic improvement had shaped such alternative fruit flavors are still not well characterized.

Recent genomic studies of cultivated peaches and some of their wild relative species have identified specific genome regions targeted by human selection, some of which are related to fruit taste flavor, clarifying that such selection occurred both during domestication[11,20] and subsequent improvement efforts[20,21]. However, much remains unknown about how specific improvement-related loci/genes have contributed to peach fruit flavor. Although previous studies in peach have reported some QTLs and/or candidate genes for fruit sweetness and acidity flavor-related traits[22–29], their actual genetic determinant(s) underlying these QTLs have not been identified. Partially accounting for difficulties in advancing from the peach QTLs down to the gene level, the resolution power for linkage studies has been restricted in peach by its narrow genetic basis and high level of linkage disequilibrium (LD)[15]. Ultimately, these are related to its long-generation time and self-compatibility[30]—few recombinant events and the small sizes of examined segregating populations for linkage analysis—as well as the relatively limited number of examined germplasm collections used in GWAS (genome-wide association study) analysis[15,27].

The current peach reference genome Lovell v2.0 (227.4 Mb, assembled based on Sanger sequencing data)[31,32] is from the doubled haploid PLOV2-2N of a western cultivar Lovell that has been widely used as rootstock[33]. Notably, the Chinese Cling peaches are regarded as the most influential germplasm in the history of global peach breeding[15,34], yet the absence of the genome assembly of this fundamental material has hindered full exploration of the genetic basis of peach improvement.

Here, we present a high-quality *P. persica* reference genome (257.2 Mb) of Longhua Shui Mi (hereafter referred to as LHSM) (Supplementary Table 1), a typical eastern "juicy honey peach" (Shui Mi Tao in Mandarin Chinese) and a representative of the Chinese Cling peaches that feature a pleasant sweet and low-acid taste flavor[35]. We also collect genome data for a total of 548 diverse *P. persica* accessions representing Chinese landraces as well as modern eastern and western cultivars, and 15 close wild relative *P. kansuensis* accessions. Population genomic analyses of these genomes identify a set of improved landraces (ILs), notable for their obviously contributions as elite germplasm for modern peach breeding worldwide, and our analyses show a clear trend of eastward dispersal of these landraces in the historical period before formal modern peach breeding was initiated. We also perform GWAS based on multi-year fruit flavor-related phenotypic data, and identify loci underlying the sweetness- and acidity-related flavor traits of peach fruits. Biochemical analyses of candidate genes using peach mesocarp tissues confirm that the *PpALMT1* (aluminum-activated malate transporter 1) promotes malate accumulation and that *PpERDL16* (early response to dehydration 6-like 16) increases fructose content during peach improvement.

## Results

**A high-quality LHSM reference genome.** The genome of LHSM was de novo assembled using 30.90 gigabases (Gb) of PacBio long reads (~120.13× coverage), 27.71 Gb of Illumina short reads (~107.73× coverage), and 37.87 Gb (~147.25× coverage) of Hi-C data (Supplementary Fig. 1 and Supplementary Table 2). Based on a *k*-mer analysis using all Illumina reads, the LHSM genome size was estimated to be ~271 Mb, with a heterozygosity of 0.32% (Supplementary Table 3). The final assembled genome size reached up to ~257.2 Mb, covering ~95% of the estimated genome (Table 1), and the assembly comprised 243 contigs with a contig N50 of 5.17 Mb. A total of 145 contigs, which accounted for 95.7% (~246.0 Mb) of the total assembled genome, were anchored into eight pseudo-chromosomes using the Hi-C reads (Fig. 1a, Supplementary Fig. 2, and Supplementary Table 4).

The LHSM genome assembly exhibited a significantly high Pearson correlation coefficient (*R*) (ranging from 0.95 to 0.99 for different chromosomes) with the recently reported peach genetic map[36] (Supplementary Fig. 3), suggesting an excellent linear agreement between the physical and the genetic map. The accuracy and completeness of the LHSM genome were supported by a high mapping rate for the Illumina reads (98.63% of 185,951,324) and the expressed sequence tags (ESTs) (94.11% of 80,805) of *P. persica* from NCBI (Supplementary Tables 5 and 6). The LHSM genome assembly exhibited a high LAI (LTR Assembly Index) score (20.67) and 97.4% (2066 out of 2121) of complete BUSCO genes could be aligned to the assembly, similar

**Table 1 Summary statistics for the LHSM genome assembly in comparison with the Lovell v2.0 reference genome.**

| Genomic feature | LHSM | Lovell v2.0 |
|---|---|---|
| Sequenced genotype | Diploid | Double haploid |
| Total assembly size | 257.2 Mb | 227.4 Mb[c] |
| Number of contigs | 243[d] | 2,525[c] |
| Largest contig | 18.8 Mb[d] | 1.5 Mb[c] |
| Contig N50 length | 5.17 Mb[d] | 255.4 kb[c] |
| Largest scaffold | – | 28.8 Mb[c] |
| Scaffold N50 | – | 7.3 Mb[c] |
| Sequences anchored to chromosomes | 246.0 Mb | 225.7 Mb[c] |
| GC content | 37.57% | 37.05% |
| Number of gaps[a] | 137 | 1,828 |
| Complete BUSCOs[b] | 97.4% | 96.8% |
| LTR assembly index, LAI | 20.67 | 21.29 |
| Repetitive sequences | 118.35 Mb/46.01% | 101.99 Mb/44.85% |
| Protein-coding genes/transcripts | 35,215/40,072 | 31,972/47,089 |
| Average transcript length | 2175 bp | 2215 bp |

[a] Gaps defined as >10 Ns.
[b] The analysis from comparisons with the eudicotyledons_odb10 database.
[c] The statistic values taken from the previous publication[32].
[d] Contigs assembled using HERA method.

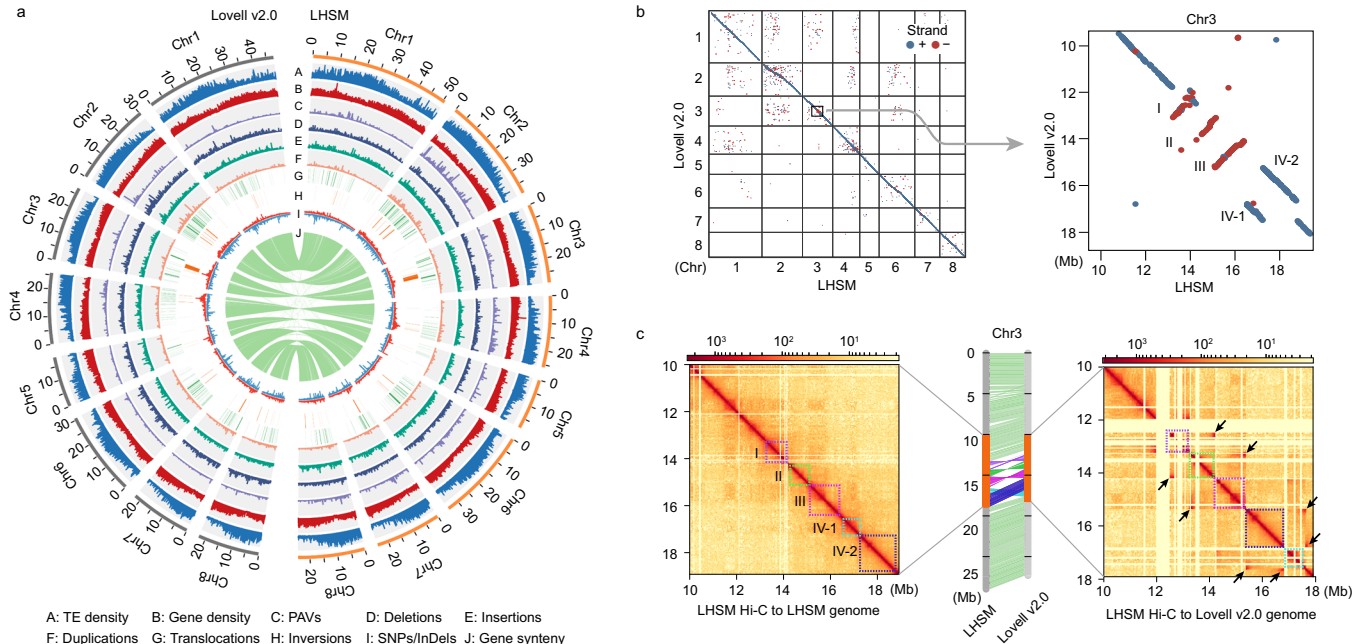

**Fig. 1 LHSM genome assembly. a** Genomic features and variation landscape across the LHSM and Lovell v2.0 genome assemblies. The outer gray (left) and yellow (right) tracks represent the chromosomes of the LHSM and Lovell v2.0 genome assemblies, respectively, with unit in Mb. **b** Genomic synteny map and three misorientations (I, II, III) and the adjacent misordering (IV-1 and IV-2) on Chr3 between these two genome assemblies. **c** Validation of the misorientations and the misordering by aligning Hi-C data of the LHSM genome to the Lovell v2.0 genome. The arrangements of I, II, III, IV-1, and IV-2 regions (highlighted by distinct color lines or boxes) on the LHSM genome are supported by contacts. The color intensity of the Hi-C heatmap represents the number of links between two 40-kb windows.

to the level obtained for the Lovell v2.0 genome (LAI: 21.29; BUSCO: 96.8%, 2054 of 2121) (Supplementary Table 7).

We predicted a total of 35,215 protein-coding genes and 40,072 transcripts (Table 1 and Supplementary Table 8), which were comparable with those of the Lovell v2.0 genome (31,972 genes and 47,089 transcripts) using the same integrative strategy combining in silico de novo gene prediction, protein-based homology searches, and transcript data from RNA sequencing analysis of various tissues (Supplementary Table 9). An analysis of TEs overlap with CDS regions indicated TEs overlap for 10,118 protein-coding genes; the percentage of CDS overlapped by TEs was 28.7% on average (Supplementary Table 10). Apart from the different methodologies used, the large difference in the number of protein-coding genes between the LHSM and Lovell v2.0 genome assemblies is likely due to a conservative selection criterion against TEs in the Lovell genome: their pipeline used an overlap value of less than 20% for TEs overlap of CDS regions[31].

The annotated protein-coding genes in the LHSM genome covered 94.1% (1996 out of 2121) of the complete BUSCO genes (Supplementary Table 7), and 88.29% of these genes could be annotated by at least one of public database (Pfam, InterPro, NR, GO, and KEGG) (Supplementary Table 11). Notably, we also annotated 118.35 Mb repetitive elements accounting for 46.01% of the LHSM assembly (Supplementary Table 12), a level slightly higher than that (44.26%) of the Lovell v2.0 genome. Collectively, these multiple lines of evidence attest to the high-quality of our de novo LHSM genome assembly, supporting its utility as an excellent reference for genomic-variation mining and genome-wide comparative analyses in peach.

We next performed analyses for genome evolution for 12 dicot plant species including seven Rosaceae (including peach) and five other species based on their 367 single-copy gene families (Supplementary Fig. 4). The maximum-likelihood phylogenetic tree revealed that *P. persica* and cultivated almond (*P. dulcis*) diverged about 4.6–16.6 MYA, consistent with the previous

reports[11,37]. We found 425 significantly expanded gene families ($P < 0.01$) comprising 4104 genes in peach as compared to the common ancestor of peach and almond (Supplementary Data 1); intriguingly, these expanded genes were significantly enriched in categories associated with defense response, ATPase activity, response to auxin, pollination, pectinesterase activity, and malate transport (Supplementary Fig. 5). Also notably, the aluminum-activated malate transporter (ALMT) gene family (gene family OG0000394 in Supplementary Data 1), which have made large contributions to fruit acidity by affecting malate content in some fruit crops, such as apple, tomato, and grapevine[5,6,38–40], was found to have higher copy number in peach (seven copies) than that (four copies) in almond.

A total of 705,879 SNPs and 181,788 InDels were identified between the LHSM and Lovell v2.0 genomes (Supplementary Table 13), potentially exerting effects on 10,234 (29.06%) protein-coding genes through the detected non-synonymous substitutions, frameshift insertions/deletions, and other large-effect mutations (stop gain, stop loss, and splicing) (Supplementary Table 14). We also identified 2309 LHSM-specific genomic segments (2.01 Mb) and 910 Lovell-specific genomic segments (0.74 Mb) (Supplementary Data 2), as well as a total of 263 LHSM-specific PAV (presence–absence variation) genes and 141 Lovell-specific PAV genes positioned within these specific segments (Supplementary Data 3). Compared with the Lovell v2.0 genome, among the syntenic regions, a total of 2653 deletions and 2068 insertions were found to affect 2.24 and 2.17 Mb genomic regions, respectively; among the rearranged regions in the LHSM genome assembly, we found 45 inversions (6.10 Mb), 391 translocations (11.22 Mb), and 1320 duplications (8.60 Mb) (Supplementary Table 15). Notably, we found a region at Chr3: 13.31—18.86 Mb, including the top-three ranked largest inversions (0.87, 0.83, and 1.27 Mb) and the adjacent translocations (0.71 and 1.53 Mb for two translocated segments); ~9% (3193) of protein-coding genes were located within or overlapped

with these InDels and rearranged regions (Supplementary Data 4 and 5). Thus, we further examined this region through comparison between the Hi-C contact matrices of the LHSM and Lovell v2.0 assemblies constructed using LHSM Hi-C data (Fig. 1b, c), and through synteny analysis between LHSM genome assembly and scaffolds of Lovell v2.0 genome. Beyond showing the complexity of this region in the Lovell genome which—was highlighted by Verde et al.[32]—these results supported the putative misordering or misorientation of some scaffolds in the corresponding region of the Lovell v2.0 genome; for example, the Super_27 and Super_451 were misordered, and their order in the pseudomolecule should be inverted in a future release (Supplementary Table 16).

In addition to variations in genomic sequences, we also explored the gene copy number variations between the LHSM and Lovell v2.0 genomes. Based on clustering analysis of orthologous genes, we found 22,166 species-conserved orthogroups covering 23,726 genes, and 2419 and 944 species-expanded orthogroups covering 7727 and 2988 genes for the LHSM and Lovell v2.0 genomes, respectively (Supplementary Table 17). GO functional enrichment analysis revealed that genes in the species-expanded orthogroups of the LHSM genome were enriched for functions related to defense response, whereas there was enrichment for genes involved in proteolysis and reproduction process in the Lovell v2.0 genome (Supplementary Fig. 6).

**Peach population structure and pre-breeding improvement in fruit quality.** We identified a total of 6.97 million SNPs and 1.23 million InDels across 548 *P. persica* genomes from various geographic regions and 15 closely wild relative (*P. kansuensis*) genomes with an average depth of 26.34× based on mapping to the LHSM reference genome (Supplementary Data 6–8). Using the *P. kansuensis* accessions as the outgroup, a neighbor-joining (NJ) phylogenetic tree for all *P. persica* accessions provided a first separation of group I (including all ornamental peaches and most of landraces) and group II (mainly including most of the modern cultivars) (Fig. 2a). Group II was further classified into two subgroups (group II-1 and II-2); group II-1 mainly contained eastern cultivars (ECs) from China and other Asian regions and group II-2 mainly contained western cultivars (WCs) notably from the Americas and Europe (Supplementary Data 6). These classifications were also supported by the principal component analysis (PCA) (Fig. 2b), the model-based clustering analysis ($K = 3$ and 4) using ADMIXTURE (Fig. 2a), and a previous study[20].

Group II-1 showed clear admixture within some ILs; another NJ-tree for all the landraces and ornamental peaches supported that these ILs from eastern China (Fig. 2c), including most of the famous Chinese Cling peaches from the Yangtze River Delta region and some elite landraces from the adjacent Huang-Huai region, are genetically derived from the primitive landraces (PLs) across western, central, and eastern China in group I. Regarding their fruit quality traits, ILs displayed remarkable improvement in higher fructose content and lower fruit acidity relative to PLs (Fig. 2d), suggesting selection of ILs by agriculturalists (an early improvement process) prior to modern peach breeding programs. A multiple sequentially Markovian coalescent (MSMC) analysis showed that PLs had an earlier expansion as well as a lager effective population size than ILs (Fig. 2e and Supplementary Fig. 7). Moreover, ILs had markedly elevated LD and reduced genetic diversity ($\theta\pi$) compared to PLs (Fig. 2f), suggesting that a bottleneck ($\theta\pi_{PL}/\theta\pi_{IL} = 1.37$) occurred during the early improvement along with the eastward dispersal. Notably, the protein-coding genes within the selective sweep regions in the comparison of ILs and PLs showed enrichment for GO terms including sucrose biosynthetic process (GO:0005986), sugar-phosphatase

activity (GO:0050308), malate metabolic process (GO:0006108), malate dehydrogenase activity (GO:0046554), organic acid biosynthetic process (GO:0016053), and regulation of pH (GO:0006885) (Supplementary Data 9 and 10), indicating the potential alteration towards fruit flavor during this early improvement process.

We compared each accession of the modern cultivars for signatures of introgressed fragments inherited from the PLs or ILs based on rIBD (relative identical by descent) analysis[41–43] (Supplementary Fig. 8a). The result indicated that the modern cultivars had larger proportions of genomic introgressions from the ILs than from the PLs. Through investigations of the genomic segments introgressed from the ILs into modern cultivars (ECs or WCs), we found genes putatively encoding enzymes or proteins known to function in the synthesis or transport of major organic acids (e.g., ALMTs[6], NADP-malic enzyme[44], isocitrate dehydrogenases[45], and H-ATPase[7]) and sugars (e.g., SWEET sugar transporters[46], tonoplast monosaccharide transporter[47], sugar transporter, polyol/monosaccharide transporter[48], sucrose synthase, phosphofructokinase[49], beta-galactosidase, and beta-glucosidases) (Supplementary Fig. 8b and Supplementary Data 11–14). These findings suggest that potential genetic source from IL peaches contributed to the fruit flavor-related traits during modern peach breeding.

**Divergent selection for fruit acidity during modern peach breeding.** Given the genetic divergence between ECs and WCs (Fig. 2a), we performed selective sweep analysis to search for the genome regions bearing strong selective signatures in the comparisons between ECs and WCs; we were also interested in identifying possible genes under selection in such regions (Fig. 3a). Notably, we found enriched GO terms related to malate (dicarboxylic acid) transport (GO: 0015743), citrate (tricarboxylic acid) metabolic process (GO: 0006101), dicarboxylic acid metabolic process (GO: 0043648), and tricarboxylic acid metabolic process (GO: 0072350) among the protein-coding genes within the selective regions, indicating potential alteration of malate and citrate accumulation (Supplementary Data 15 and 16). Moreover, we found genes encoding putative ALMT[5,6], ATP citrate lyase (ACL)[50], lactate/malate dehydrogenases (LDH/MDH)[51], isocitrate/isopropylmalate dehydrogenases (IDH/ISDH)[52], and H-ATPase[7] (Fig. 3a and Supplementary Data 17 and 18) among the enriched GO terms; homologs of these proteins have been previously implicated in the metabolism or transport of organic acids in fruit crops. These findings suggesting divergent selection for fruit-acidity-related traits during peach breeding promoted us to quantify acidity-related phenotypes in ripe fruits, including the content of the organic acids: quinic acid and shikimic acid, as well as the two major contributors for peach fruit acidity: malate and citrate[53]. We examined these phenotypes for accessions over two consecutive years (2016 and 2017), and found significantly higher levels of both malate and citrate in WCs compared to ECs (Fig. 3b and Supplementary Fig. 9). We further measured the pH of their ripe fruits, and also collected titratable acidity (TA) data for all accessions in 2017. Our phenotypic analysis for fruit acidity, as measured by TA and pH, showed that WCs have significantly higher TA level and lower pH compared to those of ECs, multiple lines of empirical evidence supporting the divergent selection for fruit acidity in ECs vs. WCs.

We noted that malate, which is the predominant organic acid in peach[18], showed the strongest correlation with pH ($R = -0.62$, $P < 0.001$ in 2017) and TA ($R = 0.76$, $P < 0.001$ in 2017) among the examined organic acids (Supplementary Data 19), apparently accounting for a large extent of the divergence in fruit acidity between ECs and WCs. Of particular note, we found that five

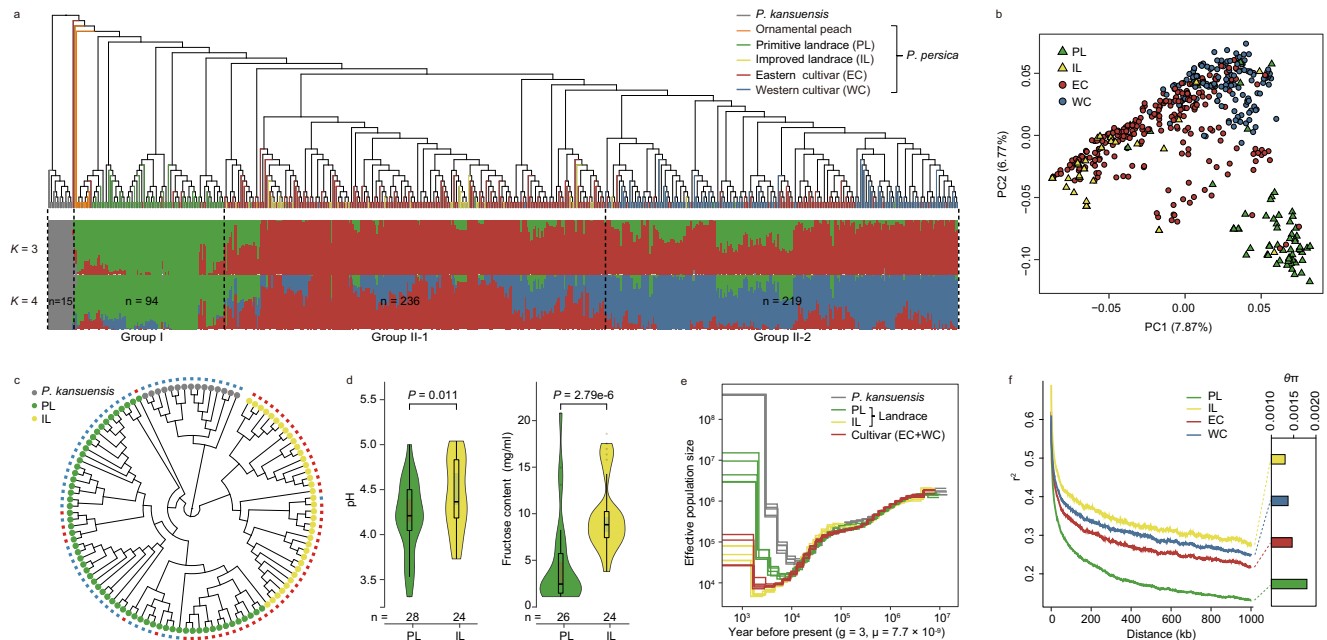

**Fig. 2 Population structure and genetic divergence of primitive landraces and improved landraces. a** Phylogenetic tree and model-based clustering of 564 peach accessions, including 549 cultivated *P. persica* accessions and 15 close wild relative *P. kansuensis* accessions. The neighbor-joining (NJ) tree and model-based clustering (*K* = 3 and 4) were constructed based on a total of 337,386 SNPs. **b** PCA plot of the first two principal components for all *P. persica* accessions (*n* = 549) supported the classification of landraces (PLs and ILs), ECs, and WCs. Accessions with ambiguous/debatable geographic origins were excluded in subsequent analyses. **c** NJ tree for all PL and IL accessions supported the genetic divergence of PLs and ILs based on the same set of SNPs. The outer squares labeled with red and blue represent the accessions from eastern China and from central and western China, respectively. **d** Significant divergence of the pH and fructose content of the PL and IL accessions. The number (*n*) of individuals for each peach type is shown below. In the violin plots (significance was tested with Wilcoxon tests), central line: median values; bounds of box: 25th and 75th percentiles; whiskers: 1.5 * IQR (IQR: the interquartile range between the 25th and 75th percentile). **e** Demographic analysis for different groupings of peach accessions. A multiple sequentially Markovian coalescent (MSMC) model was used to infer their effective population fluctuations under a mutation rate $\mu = 7.7 \times 10^{-9}$ per site per generation[112] and the generation time of 3 years (and generation time of 4 years in Supplementary Fig. 7). **f** Linkage disequilibrium (LD) patterns and genetic diversity ($\theta\pi$) for the four different types of *P. persica*. Source data underlying Fig. 2f are provided as a Source Data file.

putative ALMT encoding genes were among the genes located in selective sweep regions (Fig. 3a). We examined the expression of these ALMT genes in mesocarp tissues at 48 DAA (days after anthesis), a period corresponding to the primary phase for malate accumulation in peach based on our data (Supplementary Fig. 10) as well as a previous study[19]. One ALMT gene (*Pp.LH.06G01819*) was expressed at a significantly higher level (*P* = 0.004, two-sided Student's *t*-test) in fruits of three high-malate WC accessions compared to fruits of three low-malate EC accessions (Fig. 3c). Phylogenetic analysis showed that *Pp.LH.06G01819* was clustered into the corresponding Arabidopsis ALMT clade I with the previously reported TaALMT1, as a malate channel in wheat[54] (Supplementary Fig. 11). We named *Pp.LH.06G01819* as *PpALMT1*, and peach mesocarp tissues transiently overexpressing *PpALMT1* had significantly increased malate content compared to vector control mesocarp tissues (Fig. 3d). These results indicate that PpALMT1 functions to increase malate content in peach fruit and supports the inference that differential expression of *PpALMT1* has likely contributed to the divergence of ECs and WCs in fruit acidity during modern peach breeding.

To further explore whether the genetic loci associated with acidity have undergone divergent selection, we performed GWAS analysis of four acidity-related traits including pH, TA, malate content, and citrate content. For pH, malate, and citrate, we respectively detected 11, 8, and 4 significant loci in 2016 and 20, 11, and 3 loci in 2017, and for TA, a total of 16 significant loci were detected in 2017 (Supplementary Table 18 and Supplementary Data 20). One strongly associated locus (Chr5: 21,714–1,812,811 bp) explained a large proportion of the

phenotypic variance across these four traits (ranging from 9.43 to 38.04%) (Supplementary Fig. 12); this overlapped with the known *D* locus of chromosome 5, which has been variously reported to exert a large-effect on TA or pH[22,23,25,28,29]. In addition to this locus on chromosome 5, there were other significant loci with relatively high PVE (phenotypic variance explanation) values (6.46–27.11%), results implying a complicated genetic regulation mechanism underlying fruit acidity. In particular, a significantly associated locus (Chr2: 29,927,641 bp) was among the very top-ranking loci in terms of both *P* and PVE values for all four traits in at least 1 year, findings clearly suggesting its potential contribution to fruit acidity. It also bears mention that 26.7% to 100.0% of the peak SNPs for each trait positioned within acidity-associated loci shared overlap (or were nearby; <100 kb) with the selective sweep regions between ECs and WCs (Supplementary Table 19 and Supplementary Data 20). Beyond clearly indicating that these genetic loci have contributed to the divergent selection of fruit acidity traits between ECs and WCs, these GWAS results provide an empirical basis for investigating causal variations for fruit acidity, a major organoleptic determinant of fruit flavor quality.

**Genetic loci associated with major sugars underlying peach fruit sweetness.** Another major organoleptic aspect of fruit flavor is sweetness, which is determined by both the type and content of soluble sugars, including for example sucrose, fructose, glucose, and sorbitol[27,55]. We quantified the content of these four major sugars for the ripe fruits of the *P. persica* accessions over two

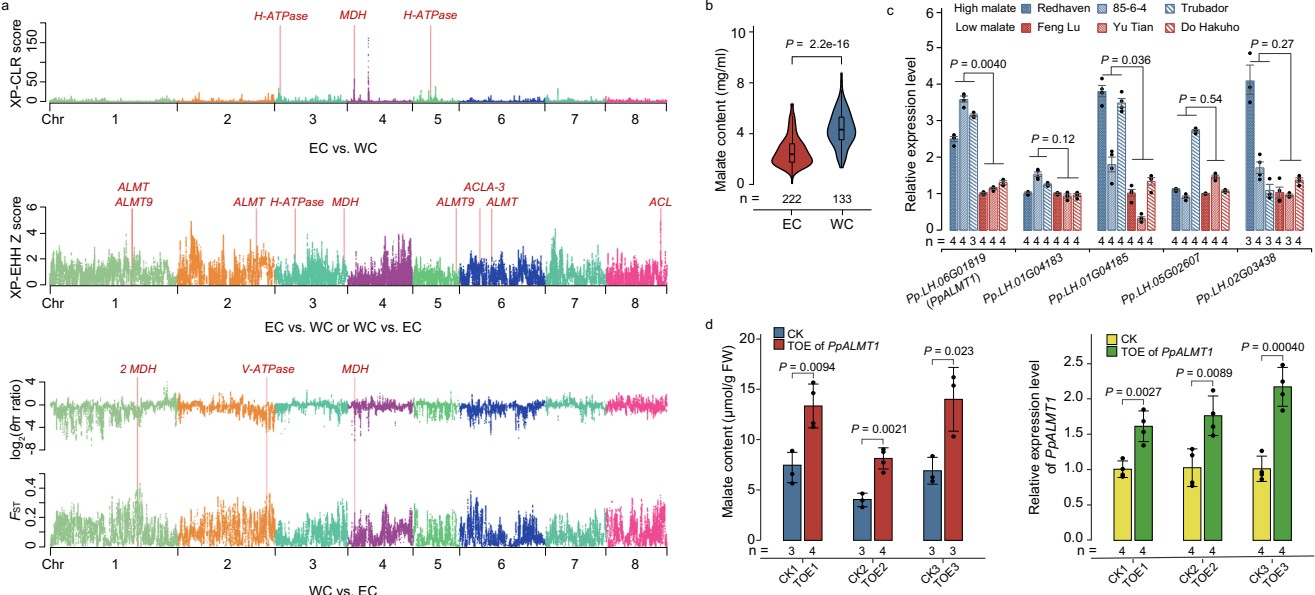

**Fig. 3 Divergent selection for fruit acidity during modern peach breeding. a** Selection signatures in the comparisons between ECs and WCs. Possible genes hypothetically under selection within the putative selective regions are labeled in red. **b** Significant divergence of the malate content between ECs ($n = 222$) and WCs ($n = 133$). In the violin plots (significance was tested with Wilcoxon tests), central line: median values; bounds of box: 25th and 75th percentiles; whiskers: 1.5 * IQR (IQR: the interquartile range between the 25th and 75th percentile). **c** Relative expression levels of the five putative ALMT genes in fruits of three low-malate EC accessions vs. three high-malate WC accessions at 48 DAA (days after anthesis). The number ($n$) of individuals for each accession is shown below. Data were presented as the mean ± SE. Significance was tested with two-sided Student's $t$-tests. **d** A significantly elevated level of malate content (left) in peach mesocarp tissues transiently over-expressing (TOE) *PpALMT1* (right). Fruit mesocarps transiently transformed with the empty vector were used as controls (CK). The number ($n$) of each independent experiment is shown below. Data were presented as the mean ± SD. Significance was tested with two-sided Student's $t$-tests. Source data underlying Fig. 3c, d are provided as a Source Data file.

consecutive years (2016 and 2017). Specifically, for 2016 and 2017 data, we found that sucrose accounts for ~76.93 and 75.10% of the examined sugar content at average, followed by glucose (8.65 and 12.16%), fructose (9.98 and 9.58%), and sorbitol (4.44 and 3.16%) (Supplementary Fig. 13), similar trends as in previous studies[27,56,57].

We performed GWAS to identify significantly associated loci for the content of these four sugars based on 1,067,831 SNPs with minor allele frequency (MAF) ≥0.05 (Supplementary Fig. 14). A major locus on chromosome 5 (Chr5: 614,754—1,109,368 bp) explained 8.6 and 7.6% of the phenotypic variance for sucrose content in 2016 and 2017, respectively (Fig. 4a), and this overlapped with previously reported QTLs for sucrose content on chromosome 5 identified by using the hybrid populations[22–24] (Supplementary Data 21). It is notable that a gene (*PpTST1*) encoding a tonoplast sugar transporter (TST) is positioned adjacent to this locus from our GWAS. TST proteins can load soluble sugars into the vacuole[58,59], and the *PpTST1* was recently reported to affect sucrose content in peach fruit[60]. We identified four loci significantly associated with glucose content in 2017 (Fig. 4b) (Chr1: 30,732,072—30,732,099 bp, Chr3: 15,707,662—15,707,662 bp, Chr4: 10,736,973—12,413,438 bp, and Chr8: 14,342,373—14,343,414 bp), respectively explaining 7.1, 6.8, 12.5, and 7.1% of the phenotypic variance for glucose content. The major locus on chromosome 4 was found to overlap with a previously reported glucose-related QTL (Supplementary Data 21). Within this major locus (Chr4: 10,736,973—12,413,438 bp), *Pp. LH.04G02050* encoding a putative β-glucosidase that catalyzes hydrolysis of β-D-glucoside or oligosaccharide substrates[61] may regulate glucose accumulation. Sorbitol is universally found in stone fruits and is a significant contributor to sweetness in peaches[19,62]. We detected two significant signals on chromosome 6 associated with sorbitol content in 2016, and one signal each for chromosome 1 and chromosome 3 in 2017 (Supplementary Fig. 14); these respectively

explained 7.5, 8.1, 7.9, and 7.7% of phenotypic variation, thus identifying candidate loci for investigations about the genetic determinants of sorbitol accumulation. Of these, one signal on chromosome 6 (Chr6: 22,350,242—22,451,210) was found to overlap with a recently reported QTL for sorbitol[63] (Supplementary Data 21).

Fructose has a higher sweetness impact (1.7-fold as compared to sucrose) compared to sucrose, glucose, or sorbitol[64]. Selection for the elevation of fructose content in tomato has been applied to develop sweeter cultivars[65,66]. In this study, we identified a major GWAS locus (Chr1: 11,738,129— 12,006,040 bp) for fructose content (Fig. 4c); this overlapped the previously reported FRU QTL on chromosome 1 identified by using a hybrid population[24] (Supplementary Data 21). The peak SNP ($P = 6.49e-16$) in the major locus could explain up to 13.87% of phenotypic variation for fructose content in our panel. It was notable that this locus showed a strong selection signature, supported by significantly reduced nucleotide diversity ($\theta\pi$) from primitive (PLs) to improved (ECs, WCs, or ILs) (Fig. 4d). This finding is particularly interesting when considering the results of our comparative sugar content analyses collectively: our data support that only fructose content has been elevated during the peach improvement (Fig. 4e and Supplementary Fig. 15). Accordingly, the raised sweetness levels of improved peach germplasm have resulted from elevated accumulation of fructose. This conclusion agreed with the previous suggestion that commercial high-quality peaches have higher fructose content as compared to native peach accessions[27,67].

## Identification of the *PpERDL16* gene and its contribution to increased fructose accumulation during peach improvement.
The haplotype blocks were estimated using PLINK in our candidate region for fructose content; this effort further narrowed this region into only two haplotype blocks harboring significantly

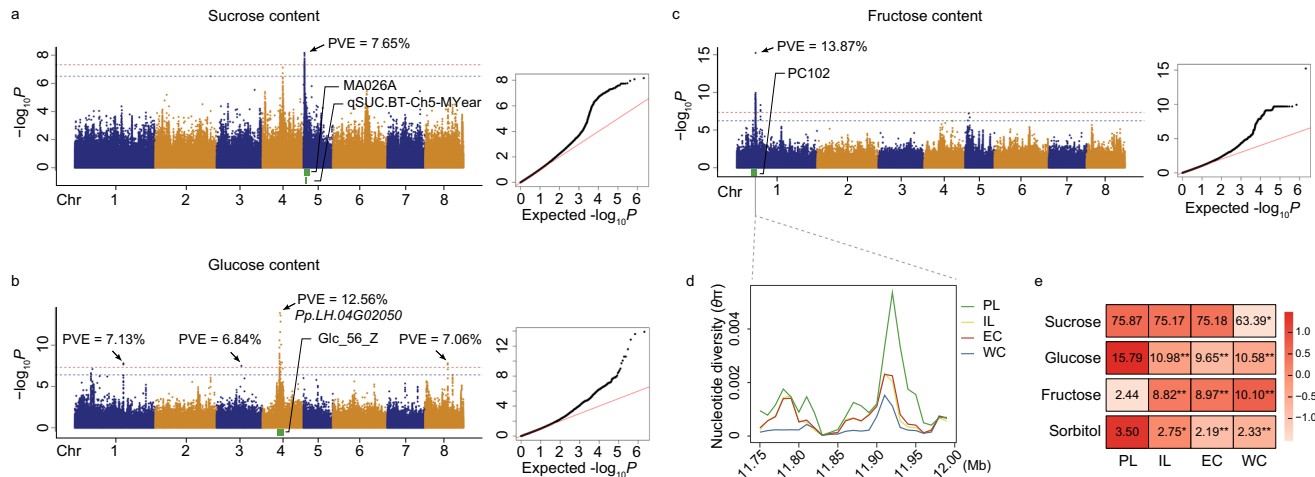

**Fig. 4 Genetic loci associated with the content of sucrose, glucose, and fructose affecting peach fruit sweetness.** GWAS results from analyses of data for sucrose (**a**), glucose (**b**), and fructose (**c**) content in 2017, respectively. The horizontal lines depict the Bonferroni-adjusted significance threshold (red) and Permutation threshold (blue) in the Manhattan plot. The indicated PVE (phenotypic variance explanation) value is for the lead SNP at the major locus for each trait. The green bar at the bottom of each Manhattan plot represents previously reported QTLs[27, 68, 129] overlapping the major loci of these three traits. **d** Reduced nucleotide diversity ($\theta\pi$) in the specific genome region, corresponding to the major locus of fructose content on Chr1, of the ILs, ECs, and WCs, as compared to PLs. **e** Representation of fruit sugar profiles in the different peach types representing the course of peach improvement (*$P < 0.05$, **$P < 0.01$ in two-sided Student's $t$-tests). Among the four examined sugars, only fructose content has been significantly elevated during peach improvement. The scaled value of sugar content is shown at the right side.

associated SNPs (block1: Chr1: 11,735,344—11,784,598 bp and block2: Chr1: 11,912,057—11,962,326 bp) (Fig. 5a). A qPCR analysis of ripening fruits showed that one gene (*Pp.LH.01G01754*) out of all the 13 protein-coding genes found within these two blocks had notably higher expression in the three tested low-fructose accessions compared to the three high-fructose accessions (Supplementary Fig. 16). We also found that its expression level was significantly negatively correlated with fruit fructose content ($R = -0.56$, $P = 3e-04$, two-sided Student's $t$-test) in a larger panel of 37 peach accessions (Fig. 5b), helping to explain the earlier report that the FRU QTL region displayed a strong negative effect on fructose content throughout fruit development[68].

Phylogenetic analysis showed that Pp.LH.01G01754 belongs to the ERD6-like subfamily of monosaccharide transporters and it has the closest relationship with ERD6-like 16 (early response to dehydration 6-like 16) protein of Arabidopsis (Supplementary Fig. 17), so it was designated as *PpERDL16*. Previous studies showed that AtERDL6 in Arabidopsis and MdERDL6-1 in apple are symporter proteins that function in glucose export from the vacuole into the cytosol[69,70], and transgenic Arabidopsis lines overexpressing *AtERDL6* showed lower levels of glucose and fructose in leaves as compared to wild type plants[69]. We confirmed the tonoplast localization of the PpERDL16-GFP fusion protein in tobacco leaf cells using the Atγ-TIP-mCherry fusion protein as the positive control (Fig. 5c). We also examined peach mesocarp tissues transiently overexpressing *PpERDL16*, and found that mesocarp tissues infiltrated with the *PpERDL16* vector had significantly reduced levels of both glucose and fructose compared to empty vector control mesocarp tissues (Fig. 5d). Viewed collectively, these results support that *PpERDL16* is very likely the causal gene underlying the previously reported major FRU QTL locus for fruit fructose accumulation.

We found that the $\theta\pi$ values of *PpERDL16* (both its CDS and the upstream (~5 kb) region harboring potential *cis*-regulatory elements) were lower among the modern cultivars (ECs or WCs) compared to PLs (Supplementary Fig. 18), which could hypothetically have resulted from selection for *PpERDL16*. This motivated an additional detailed analysis of the *PpERDL16*

throughout peach improvement. After filtering 17 low frequency haplotypes (i.e., only carried by one accession), all 76 SNPs in the genic region of *PpERDL16* could be classified into eight haplotypes for all peach accessions (including 15 *P. kansuensis* accessions) (Fig. 5e). Haplotype network analysis showed that the primitive haplotypes (Hap6–8) were only carried by wild relative *P. kansuensis* (Fig. 5e and Supplementary Fig. 19), whereas Hap1–5 occurred in ornamental peaches, peach landraces, and cultivars, with the highest frequency (86.5%) for Hap4, followed by Hap1 (8.7%), Hap5 (5.4%), Hap2 (1.7%), and Hap3 (0.2%) (Fig. 5e). Moreover, the fructose content of the accessions carrying Hap4 (average 9.58 mg/ml) or Hap5 (average 8.61 mg/ml) was significantly higher than those carrying haplotype Hap1 (average 2.39 mg/ml), Hap2 (average 3.89 mg/ml), and Hap3 (average 2.39 mg/ml) (Fig. 5f). The frequencies of the Hap4 and Hap5 were increased among ILs, ECs, and WCs, as compared to ornamental peaches and PLs (Fig. 5g). Finally, consistent with our speculations about PpERDL16's function, the result that Hap4 was carried by 92.8% of ECs and 98.3% of WCs highlighted apparently convergent selection for increased fructose content in both ECs and WCs.

## Discussion

We present a high-quality LHSM reference genome and mapped resequencing data for a large natural population comprising 564 peach accessions to this genome. These resources collectively explain the extent of genetic variations in peach substantially, thereby supporting peach genetic studies by augmenting resolution power for association studies. This is significant, because the resolution power has long been dragged down owing to the narrow genetic basis and high levels of LD in peach[15,27]. It is notable that our study revealed a historical eastward dispersal and continuous improvement trend for domesticated peaches that occurred before modern peach breeding efforts. It was these early efforts which led to the low-acid and sweet ILs, including the typical Chinese Cling peaches, that have subsequently served as elite germplasm, a situation reflected in the overwhelming contribution of the ILs to the modern cultivars as compared to PLs.

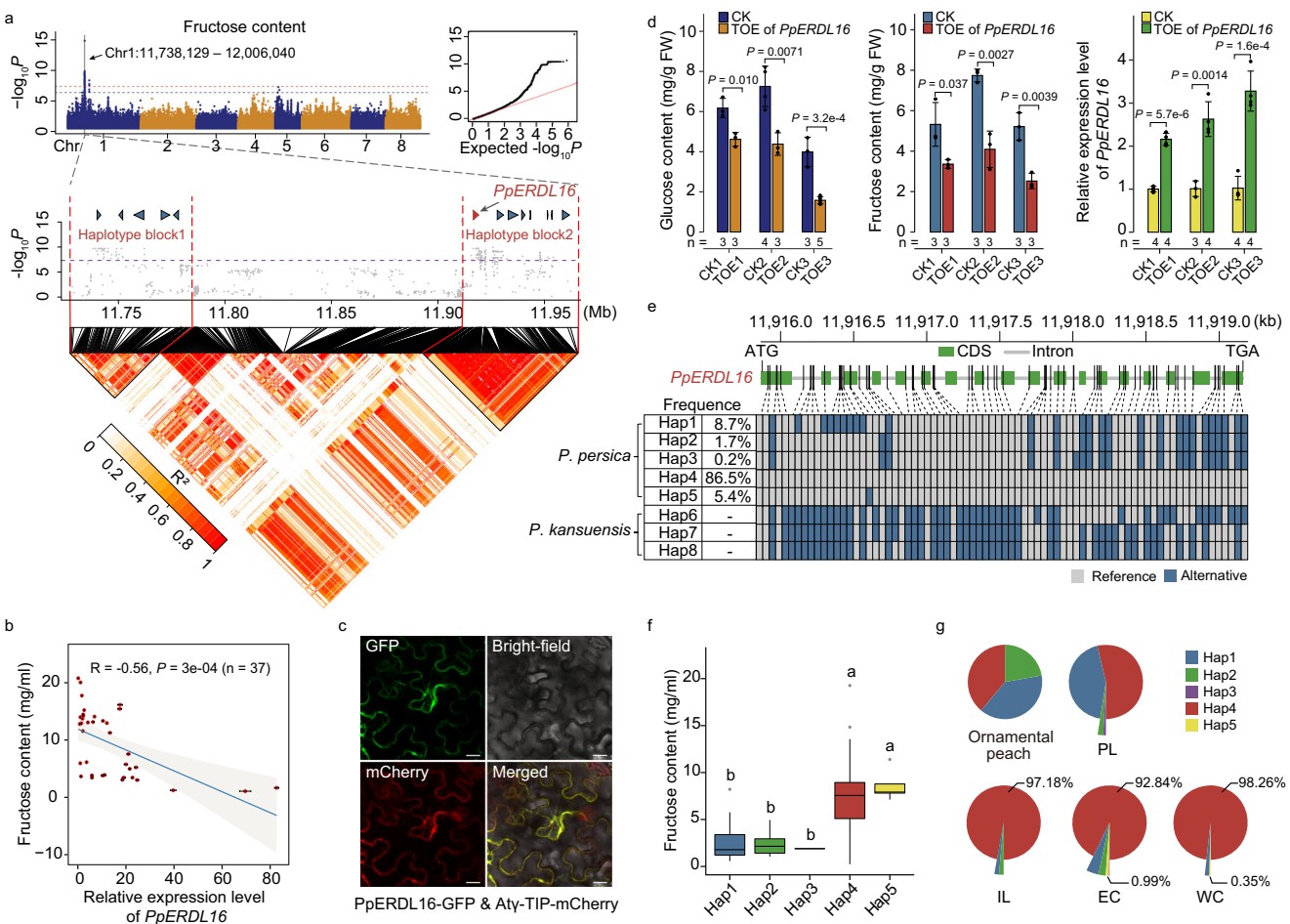

**Fig. 5 Identification of *PpERDL16* and its contribution to increased fructose accumulation during peach improvement. a** Two significantly associated haplotype blocks of the major GWAS locus for fructose content. The horizontal lines depict the Bonferroni-adjusted significance threshold (red) and Permutation threshold (blue) in the Manhattan plot (top). The two significantly associated haplotype blocks are represented by red triangles surrounding black lines in the heatmap (bottom). The 13 protein-coding genes (including *PpERDL16* in red) in these two haplotype blocks are shown as colored arrows. **b** A significantly negative correlation between fructose content and the *PpERDL16* expression level in 37 peach accessions. Data were presented as the mean ± SD. Significance was tested with two-sided Student's *t*-tests. **c** Confocal imaging of tonoplast localization of the transiently expressed PpERDL16-GFP fusion protein in tobacco leaf cells using the Atγ-TIP-mCherry fusion protein as the positive control. Three independent experiments were performed with similar results. Scale bar, 20 μm. **d** Significantly reduced levels of both the glucose (left) and fructose (middle) content in peach mesocarp tissues transiently overexpressing (TOE) *PpERDL16* (right). Fruit mesocarps transiently transformed with the empty vector were used as controls (CK). FW fresh weight. The number (*n*) of each independent experiment are shown below. Data were presented as the mean ± SD. Significance was tested with two-sided Student's *t*-tests. **e** Gene structure and haplotypes of *PpERDL16* among peach (*P. persica*) accessions, as well as *P. kansuensis* accessions. The frequency for each haplotype is calculated for the *P. persica* accessions. **f** The distribution of the fructose content of each haplotype type was displayed in the box plot. Multiple comparison was conducted using least sgnificant difference (LSD) test. In the box plots, central line: median values; bounds of box: 25th and 75th percentiles; whiskers: 1.5 * IQR (IQR: the interquartile range between the 25th and 75th percentile). **g** The frequency distribution of *PpERDL16* haplotypes (Hap1–5) in different types of *P. persica* accessions (ornamental peaches, PLs, ILs, ECs, and WCs). The different colored portions in each pie chart represent the percentage of different haplotypes. The percentage values of Hap4 and Hap5 exhibiting higher fructose content were displayed. Source data underlying Fig. 5b–g are provided as a Source Data file.

Nevertheless, our data also show that the PLs have much higher genetic diversity than the ILs, supporting their utility for breeding and improvement applications requiring an expanded genetic basis for introducing economically important traits into modern peaches (e.g., the potential for resistance to viral pathogens, etc.). Additionally, our LHSM genome, as a typical IL genome, will surely facilitate mining of valuable genomic information for peach genetic improvement generally and specifically for efforts to modulate fruit quality traits.

Eating quality is an important aspect for the improvement of fruit-bearing crops as well as seed crops like rice, maize, wheat, and soybean[71]. Sweetness and acidity are understood as the two most impactful organoleptic attributes for fruit flavor. In peach,

the common consumer demand for sweeter taste, coupled with differentiated cultural preferences for acidity, has resulted in the formation of two typical flavor types: sweet, low-acid taste vs. sweet, acid taste, respectively favored by eastern and western consumers[18,19]. Similar preferences are evident for apple cultivars: North Americans and Europeans favor sweet, sub-acid apples, whereas sweet apples with barely any acid flavor are preferred in Asia and India[72]. Our data revealed signatures of selection in the peach genome that underlie the divergent selection for fruit acidity that has occurred between eastern and western peach breeding programs. And we used these detected differences to pursue specific acidity-related loci and/or candidate genes. Among candidate genes, a *PpAMLT1* gene was found to

affect accumulation of malate, the predominant acid in peach fruits, thus illustrating the utility of our data and providing specific information to support genetic improvement towards acidity. Despite the dominant role of malate in contributing to the divergence of fruit acidity between ECs and WCs, it is bears mention that the significantly increased citrate content in WCs, as compared to the PLs (Supplementary Fig. 9), could also serve as a non-ignorable factor in elevating the fruit acidity in WCs.

We show that *PpERDL16* is a casual gene that controls fructose accumulation in peach fruit and haplotype analysis clearly highlighted how this locus has driven the elevation of sweetness that has advanced during multiple stages of peach improvement. Our study also provides an excellent example for how a single phenotype (sweetness) desired by consumers can be obtained via separate selection trajectories in multiple fruit crop species involving distinct biochemical mechanisms. For example, the selection of *ClTST2* which encodes a TST, led to the increased accumulation of sucrose and hexoses in watermelon[47], whereas the increased sweetness in peach and some tomato varieties[66] result from elevated fructose content as controlled by *PpERDL16* and *SlFgr* (encoding a tomato SWEET transporter), respectively. More broadly, the differentiation of maize into field corn and sweet corn varieties resulted from altered starch biosynthesis as mediated by a mutation in *ZmSUGARY1* (encoding an isoamylase-type starch-debranching enzyme)[73].

In summary, our study shows how harnessing a high-quality genome assembly for a long-prized improved Chinese landrace ultimately supported development of additional genome-scale germplasm diversity resources at a population scale. Beyond providing valuable genomic resources for peach genomic and genetic research, our study provide insight into the improvement of peach flavor, revealing genetic basis underlying fruit flavor. Our findings also provide a genomic framework for fruit crops that can deepen understanding of fruit quality trait physiology and that suggests strategies for flavor improvement.

## Methods

**Plant materials**. The sequenced peach (*P. persica*) accessions used in this study were obtained from the Beijing and the Nanjing National Peach Germplasm Repositories, China. The 15 *P. kansuensis* accessions were collected in the Gansu province of China. A representative Chinese Cling peach (cv. Longhua Shui Mi (LHSM)) was collected from the Nanjing National Peach Germplasm Repository, China.

**DNA extraction and sequencing**. Extraction and purification of high molecular weight DNA was performed using the DNeasy Plant Maxi Kit (Qiagen, Germany). DNA concentration was measured using a NanoDrop spectrophotometer (Thermo Fisher Scientific, USA) and the Qubit 2.0 Fluorometer (Invitrogen, USA). Illumina short-read data were obtained using the Illumina NovaSeq platform, which generated a total of 184.73 million reads with a total length of 27.71 Gb. Single-molecule real-time (SMRT) cells were sequenced on the PacBio Sequel platform (Pacific Biosciences, CA, USA), generating a total of 3.54 million reads with a total length of 30.90 Gb. Hi-C libraries were created from young leaves, which were fixed with formaldehyde and then lysed before the cross-linked DNA was digested overnight with *Dpn*II. Sticky ends were biotinylated and proximity-ligated to form chimeric junctions that were enriched for, and then physically sheared to a size of 500−700 bp. Chimeric fragments representing the original cross-linked long-distance physical interactions were processed into paired-end sequencing libraries. This allowed us to generate a total of 126.25 million paired-end reads and 37.87 Gb of sequencing data on an Illumina NovaSeq platform. The alignment of the Hi-C reads was implemented using the HiC-Pro program[74] and revealed a high proportion (82%) of valid interactions that confirmed the high quality of the Hi-C data (Supplementary Fig. 20).

**Genome assembly**. In order to estimate the genome size of LHSM, the Illumina short reads were recruited to determine the *k*-mer distributions using the GenomeScope software[75]. The PacBio long-read data were de novo assembled into PacBio contigs using Canu version 1.9[76], generating a total of 2212 contigs with a N50 of 686.03 kb. We then used the Highly Efficient Repeat Assembly (HERA) method[77] based on the Canu-corrected PacBio long-read data in order to extend the PacBio contigs to 243 contigs (HERA contigs v1) with a N50 of 5.17 Mb. The

Illumina short-read data were used for error correcting the contigs using Pilon[78]. Subsequently, and in order to anchor the corrected contigs (HERA contigs v2) into chromosomes, we aligned the Hi-C sequencing data into these contigs using Juicer v1.8.9[79]. The contigs were finally linked into eight distinct chromosomes by 3D-DNA[80].

**Repeats and gene annotation**. The annotation of transposable elements was performed using RepeatMasker (http://www.repeatmasker.org). The repeat libraries included the RepBase-20170127 and the de novo repeat library created using RepeatModeler (http://www.repeatmasker.org) (with the parameter -LTRStruct). The LTRharvest[81] and the LTR_FINDER[82] programs were used to identify intact LTRs in the genome assembly and to calculate the LAI index[83].

The pipeline for ab initio gene annotations included de novo gene predictions of the repeat-masked genome using AUGUSTUS[84] and SNAP[85], as well as evidence-based gene annotations using MAKER2[86]. For de novo gene prediction, we used the AUGUSTUS and SNAP programs trained on the homolog protein-coding genes of *Arabidopsis thaliana*, *Oryza sativa*, and *P. persica*. The homolog sequences were collected from the Swiss-Prot database. Transcript evidence included transcripts assembled from RNA-Seq data obtained from different tissues (root, leaf, flower stages, and fruit; see Supplementary Table 9) using HISAT and StringTie[87]. This evidence was submitted to MAKER2, and the output was refined by the AED metric (AED <0.7). Gene functional annotation was achieved using BLASTP (−evalue < 1e − 5) against the Swiss-Prot, Pfam[88], and the NR databases[89], as well as using InterProScan version 5.27-66.0[90] against the InterPro database[91]. Gene Ontology terms were obtained for each gene from the corresponding InterPro entries. The pathways associated with each gene assigned by BLASTP[92] against the KEGG database[93], with an E-value cut-off of 1e − 5.

**Evaluation of genome assembly**. The flanking sequences of the molecular markers obtained from the high-density and the multi-population consensus genetic linkage map for peach[36] were mapped against the LHSM genome assembly using BLASTN. The Pearson correlation coefficient was computed between the genetic distance and the physical position of the uniquely aligned markers. The Illumina short-read data were also used to evaluate assembly accuracy and completeness using BWA-MEM version 0.7.17-r1188[94]. The completeness of the genome assembly and the gene annotations were assessed with a plant database composed by 2121 conserved plant genes (eudicotyledons_odb10) using BUSCO version 3.0.2[95]. The EST sequences that were retrieved from NCBI were aligned to the genome assembly using GMAP (version 2019-09-12)[96].

**Gene families and phylogenetic analysis**. We used OrthoFinder (v2.3.9)[97] to identify shared gene families between peach and 12 other plant species, including six Rosaceae (almond, apricot, European pear, apple, black raspberry, and woodland strawberry), one Brassicaceae (Arabidopsis), one Rutaceae (orange), one Salicaceae (*Populus trichocarpa*), one Vitaceae (grape), one Solanaceae (tomato), and one monocot (rice). Based on the protein sequences of 367 single-copy ortholog families, the phylogenetic relationship among these species was estimated using RAxML (v8.2.12)[98]. Divergence times were estimated by the MCMCtree program embedded in PAML (v4.9)[99]. We measured the expansion and contraction of orthologous gene families based on the maximum likelihood tree using the software CAFE v4.2 (https://github.com/hahnlab/CAFE).

**Comparative genomics**. Genome alignment between LHSM and Lovell v2.0 was performed using the NUCmer program embedded in MUMmer[100] with the parameters "-mumreference -g 1000 -c 90 -l 40". The delta-filter program was used to remove the mapping noise and to determine the one-to-one alignment blocks with parameters "-r -q". SNPs and InDels were identified using the show-snps program (-ClrT -x 1). Gene synteny analysis was performed using the MCScanX package[101] and BLASTP with the parameters "-evalue < 1e-10, -v 5, -b 5" in order to determine the pairwise similarity between the protein sequences of the LHSM and the Lovell v2.0 genomes.

To identify the presence/absence variations (PAVs) in the LHSM genome, we divided it into 500 bp overlapping windows with a step size of 100 bp. Each 500 bp window was then aligned against the Lovell v2.0 genome using BWA-MEM with the parameters "-w 500 −M". The genetic sequences within the different windows that failed to align with the Lovell v2.0 genome, or those that aligned with less than 25% coverage, were defined as LHSM-specific sequences. Overlapping windows that could not be aligned were merged together. The Lovell-specific sequences were then identified following the same method.

In order to identify structural rearrangements, we used Minimap2 v2.17-r941[102] to align the LHSM assembly to the reference Lovell v2.0 genome with the following parameter setting "-ax asm5 −eqx". Structural rearrangements and local variants (>50 bp) were detected using SyRI[103]. To identify gene copy number variation, we first performed the gene family clustering using OrthoFinder version 2.3.9[97] based on the protein sequences from the LHSM and the Lovell v2.0 genomes, and identified CNVs using a PERL script developed in-house.

**SNP and small InDel calling**. We collected Illumina resequencing data for 564 peach accessions (Supplementary Data 6) with an average depth of 26.34×. These

included 379 newly sequenced accessions. The quality control for the raw resequencing data was performed using fastp version 0.20.1[104] with default settings. For SNP calling, Illumina short reads were aligned to the LHSM genome using BWA-MEM; PCR duplicates were removed using Picard version 1.118 (http://broadinstitute.github.io/picard/). SNPs and InDels were identified using HaplotypeCaller available from the Genome Analysis Toolkit (GATK, version 4.1.5.0)[105], and subsequently filtered following ref. [3]. SNPs with a read depth <5 and non-biallelic SNPs were removed from further analyses.

**Phylogenetic and population structure analyses.** A total of 337,386 SNPs with a MAF ≥0.05, missing rate ≤50%, and with a Hardy–Weinberg Equilibrium (HWE) $P$ value >1e-6 was used to build a maximum likelihood phylogenetic tree, as well as to perform population structure and PCA. The phylogenetic tree was built using the FastTree2 program (version 2.1.10)[106]. Population structure was investigated using ADMIXTURE[107] and evaluating each $K$ from 2 to 12. PCA was performed using the smartPCA program embedded in the Eigensoft package version 7.2.1[108].

**Relative IBD and introgression analysis.** To investigate introgression from the PLs and ILs to each accession of the modern peach cultivars, we performed pairwise IBD analysis by first phasing the genotypes using Beagle (v5.1)[109] and then detecting shared IBDs tracks between any two accessions using RefinedIBD (v17Jan20.102)[110]. After this, we counted the number of shared IBD tracks in 10-kb sliding windows (in steps of 5-kb) between each modern cultivar and PLs or ILs. These counts were then normalized as nIBD = shared IBD number/number of PLs or ILs), and the rIBD was calculated as rIBD = nIBD$_{IL}$ – nIBD$_{PL}$. Average rIBD values of individuals in ECs or WCs were calculated along each window and then normalized following a standard normal distribution. Windows with $Z$-scores greater than 2 were considered as putative introgressed regions.

**Multiple sequentially Markovian coalescent analysis.** MSMC2 (v2.1.1)[111] was used to infer the demographic history of peach. To improve reliability, genome regions were masked with SNPable tool (http://lh3lh3.users.sourceforge.net/snpable.shtml) when the coverage depth was <15× after removing reads with mapping quality <20. First, we split the reference genome into overlapping 35-mers and then mapped these back to the reference genome using BWA (bwa aln -R 1000000 -O 3 -E 3). Only regions where the majority of 35-mers were uniquely mapped and without mismatch were retained for further analysis. We selected the top ten samples in each population with the highest coverage after masking. The eight most frequent haplotypes were randomly selected from the ten samples in order to infer the demographic history of each population. We repeated this procedure 20 times. Scaled times were converted to years by assuming a generation time of 3 and 4 years, respectively and a mutation rate of $7.7 \times 10^{-9}$ per site per generation for peach, following Xie et al.[112].

**Linkage disequilibrium.** To estimate and compare the patterns of LD decay in each population, we computed the mean squared correlation coefficient ($r^2$) values between any two SNPs within 500 kb using the software PopLDdecay (v3.41)[113]. To eliminate the potential effects of sample size, we randomly sampled ten accessions for each population (we repeated this procedure 100 times). We used a 500 bp bin size to generate the plot.

**Genetic diversity.** Genetic nucleotide diversity ($\theta\pi$, the average number of pairwise nucleotide differences per site between any two randomly chosen DNA sequences from the population) was calculated using VCFtools (v0.1.17)[114] on 20 kb sliding windows (with a step size of 10 kb) across the peach genome.

**Selective sweeps.** We used multiple methods to detect regions and genes under positive selection. SNPs with MAF below 5% were removed from this analysis. To identify potential selective sweeps between population A and population B, $\log_2(\pi_B/\pi_A)$ and $F_{ST}$ was calculated together using VCFtools (v0.1.17)[114] on a 20 kb sliding window with step size of 10 kb. Windows that contained less than ten SNPs were excluded from further analysis. The windows that were simultaneously (1) in the top 5% of Z-transformed $F_{ST}$ values and (2) in the bottom 5% of $\log_2 (\pi_B/\pi_A)$ were considered as candidate selective regions in population A. XP-CLR[115] is a method that uses allele frequency differentiation at linked loci between two populations to detect selective sweeps. Each chromosome was analyzed using the XP-CLR (v1.0) program with parameters "-w1 0.0005 200 200 1 -p1 0.9". The average XP-CLR scores were calculated for each 20 kb sliding window with a step size of 2 kb. The windows in the top 1% of the XP-CLR scores were considered as candidate selective regions. XP-EHH[116] was implemented using the program Selscan (v1.1.0)[117]. The results were normalized on a 20 kb window basis and the ratio of extreme scores (|score| ≥ 2) were calculated in each window. The top 1% of windows (with the highest ratio of extreme scores) were considered as candidate selective regions. Subsequently, the results from each of the above methods were combined. The genes contained within the merged candidate selective regions along the peach genome were considered as candidate selective genes.

**GO enrichment.** R package ClusterProfiler (v3.18.0)[118] was used to perform GO enrichment analysis. The GO terms showing a $P$ value < 0.05 were considered as significantly enriched.

**Phenotypic analysis for fruit flavor related traits.** We harvested ten matured peach fruits per plant and prepared the crushed mixed fruit juice for phenotypic analysis. SSC, pH, sugar (sucrose, fructose, glucose, and sorbitol), and organic acid (malate, citrate, quinate, and shikimate) contents were measured in 2016 and 2017, and the TA was measured in 2017. The pH was measured using a pH electrode (Sartorius, PB-10). The TA was measured by titrating 25 ml of fruit juice with 0.1 mol/L NaOH to a pH = 8.1, according to "Fruit and vegetable products—Determination of titratable acidity" (GB/T 12456, 2008)[119]. High performance liquid chromatography (HPLC) was used to determine the sugar and organic acid contents following Filip et al.[120]. The fruit juice was mixed with ethanol (in a proportion of 3:7 (v/v)) prior to centrifugation at 8050 × $g$ for 5 min. The resulting supernatant was forced through PVDF 0.22-μm syringe filters and then injected into the HPLC system (LC-20A, Shimadzu). The organic acid contents were detected using a photo diode array detector (SPD-M20A) and an InertSustain C18 column (250 mm × 4.6 mm ID, 5 μm, GL Sciences Inc.). The samples were eluted with 20 mM monopotassium phosphate (KH$_2$PO$_4$, pH = 2.6) at 40 °C and injected at a flow rate of 1 mL/min. The eluted compounds were detected by UV absorbance at 210 nm. The sugars were detected using a refractive index detector (RID-10A) and Luna® 5um NH2 100 Å column (250 mm × 4.6 mm, Phenomenex). The mobile phase was 80% acetonitrile with a flow rate of 3 mL/min for peak separation at 40 °C. Organic acids and sugar contents were calculated from calibration curves obtained from the corresponding external standards.

**Transiently overexpression assay.** Transient overexpression analysis in peach mesocarps was performed following previously described procedures[121]. Briefly, the two pairs of primers (see Supplementary Table 20) were designed to amplify the full-length coding sequence of *PpALMT1* and *PpERDL16* and the PCR products were then inserted into a pGreen0029 62-SK vector. The recombinant constructs and the vector control were then chemically transformed into *Agrobacterium tumefaciens* GV3101 (pSoup). The flesh slices were taken from the peel-off mesocarps and then precultured on a MS medium at 24 °C for 24 h. The flesh slices were submerged in an *A. tumefaciens* suspension and subjected to vacuum conditions (−70 kPa). After vacuum infiltration, the flesh slices were rinsed with sterile water and cultured on a MS medium in a growth chamber (24 °C, RH 85%) for 48 h. The flesh slices were then used for phenotypic and gene expression analyses.

**GWAS analysis.** We retained peach SNPs with a MAF ≥0.05 and a missing rate ≤50% to perform the GWAS analysis. After imputation using Beagle (v4.1)[109] with default parameters, the GWAS analysis was performed based on a linear mixed model using the program Fast-LMM v2.06.20130802[122]. The $P$ value threshold for significance was estimated as 0.05/n (where $n$ corresponds to the SNP number). The phenotypic variance that was explained by each SNP was estimated[123]. The haplotype blocks were estimated using the default parameter (–hap) in Plink v1.90b6.10[124].

**Validation and quantification of gene expression.** qRT-PCR analysis was used to quantify the expression levels of the 13 candidate genes within the two significantly associated haplotype blocks from six peach accessions (three with the highest fructose content and three with the lowest fructose content as measured in 2017). A total of 37 peach accessions were used to quantify the expression of *PpERDL16*. Total RNA was extracted from the mesocarp of pre-ripened fruits using the Trelief™ RNAprep Pure Plant Kit (polysaccharides and polyphenolics-rich) (Tsingke, China). The first-strand cDNA was synthesized using a PrimeScript™ RT Reagent Kit with gDNA Eraser (Takara, Japan). Quantitative PCR was performed using the TSINGKE Master qPCR Mix (SYBR Green I with UDG) (Tsingke, China), on a StepOnePlus™ Real-Time PCR System (Applied Biosystems, USA) following the manufacturer's instructions. cDNA transcript levels were normalized to those of the reference gene *actin* using the 2$^{-\Delta\Delta CT}$ method[125,126]. The entire set of primers (see Supplementary Table 20) was designed to span an intron in order to avoid the amplification of genomic DNA. PCR reactions were performed in triplicate for each biological replicate; three or more biological replicates were used in all of the PCR reactions.

**Analysis of the subcellular localization of PpERDL16.** The *Atγ-TIP* coding region lacking the stop codon (At2g36830), which encodes a vacuolar membrane protein[127], was synthesized and cloned into the pMD85-mcherry between the CaMV35S promoter and the *mCherry* coding sequence in order to generate the 35S-γ-TIP-mCherry construct. The *PpERDL16* full-length CDS lacking the stop codon was amplified from the cDNA of Redhaven fruit using PF2 primer pairs and subsequently introduced into the pMD85-GFP vector. The resulting fusion vector pMD85-PpERDL16-GFP was co-transformed with pMD85-γ-TIP-mCherry into tobacco (*Nicotiana benthamiana*) leaves via *A. tumefaciens* (strain EHA105). The infected tissues were analyzed under a fluorescence microscope (A1R; Nikon, Japan) 72 h after infiltration.

**Haplotype analysis and median-joint network**. *PpERDL16* haplotypes were constructed using the entire set of SNPs present in the gene. SNPs were phased using Beagle (v5.1)[109]. Haplotypes with frequency less than 2 were removed. Median Joining Networks for the *PpERDL16* haplotypes were built using PopART (v1.7.1)[128] with default parameters.

**Reporting Summary**. Further information on research design is available in the Nature Research Reporting Summary linked to this article.

## Data availability

Data supporting the findings of this work are available within the paper and its Supplementary Information files. A reporting summary for this Article is available as a Supplementary Information file. The datasets and plant materials generated and analyzed during the current study are available from the corresponding author upon request. The raw resequencing data have been deposited in the Sequence Read Archive of the National Center for Biotechnology Information (NCBI) under BioProjects PRJNA715782 and PRJNA663114. The genome assembly has been deposited at GenBank under the accession JAGEPH000000000. The raw PacBio data and Hi-C data are available in the NCBI Sequence Read Archive under BioProject PRJNA707388. Online tools used in this paper include: Pfam [http://pfam.xfam.org/], InterPro [https://www.ebi.ac.uk/interpro], NR [https://www.ncbi.nlm.nih.gov/refseq/about/nonredundantproteins/], GO [http://geneontology.org], KEGG [https://www.genome.jp/kegg/]. Source data are provided with this paper.

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

## Acknowledgements

This research was supported by the National Key Research and Development Program (grant no. 2018YFD1000200) and the Financial Special Foundation (grant no. KJCX201907-2), the Innovation Capacity Building Foundation (grant no. KJCX20210432), and the Youth Foundation (grant no. QNJJ202120) from Beijing Academy of Agriculture and Forestry Sciences.

## Author contributions

H.X. and J.W. designed the research. Q.J., F.R., J.Y., J.G., Z.S., and J.Z. provided materials and information. J.G., Y.Y., Z.Z., and J.F. performed data analyses. Y.X. performed experiments and drafted related methods; H.X., Y.Y., and J.G. wrote and revised the manuscript with input and comments from the other authors.

## Competing interests

The authors declare no competing interests.
