## [Peer Review File · Nature Communications]

REVIEWER COMMENTS

Reviewer #1 (Remarks to the Author):

The authors describe a high quality peach reference genome, and the alignment of Illumina sequence to the reference. They identify genomic regions and genes correlated with sugar and acid content. They further showed the function of two proteins in sugar or acid accumulation in peach. Overall, the manuscript is well written and the results interesting to a wide audience. A few minor criticisms are listed below.

1. Define enzyme names in the abstract and line 82.
2. Lines 36-37: "Fruit flavor quality is largely determined by the sweetness and acidity which are the most important factors influencing consumer preference and acceptability." Although sweetness and acidity are important, flavor is much more complex than just sugars and acids.
3. At least in the USA, of oriental and occidental are no longer used as descriptors of the East and West.
4. Sorbitol is found in a limited number of related stone fruits and is a significant contributor to sweetness in peaches, yet it is not focused on in the paper. GWAS analysis of sorbitol may lead to a better understanding of this sugar's contribution to flavor.
5. More references are needed in the methods section, for example Interpro, KEGG, BlastP and other databases.
6. Line 620: I think a reference is added to the text here.
7. Line 626: remove highest
8. Line 636: how many biological replicates?
9. Line 642: Is it Redhaven?
10. Table 1. Should it say Average transcript length.

Reviewer #2 (Remarks to the Author):

Yu et al. drafted a new genome sequences with a landrace which is nested to representative commercial accessions (like Chinese Cling peach), and then utilized that for extensive population genetics with resequencing data from >500 accessions. They properly conducted selective sweep analyses with site frequency spectrum (SFS) and extended haplotype homozygosity (EHH)-based methods to identify the loci putatively under recent artificial selections in differentiation to geographical distribution. Furthermore, they applied extensive GWAS mainly to identify fruit quality-related loci, focusing acidity and sugar content. Lastly, they could identify an important gene undergoing artificial selection, named PpERDL16, of which expression level negatively contributes to fructose content in fruit. Overall, massive data was provided, including new draft genome sequences and >500 resequencing data. They looked properly interpreted by multiple aspects of population genetic approaches. Methods sound well. I highly appreciate the whole-genome sequencing of Chinese Cling-like accession, because the current standard, cv. Lovell peach, would be quite far from the modern cultivated peaches. Although some of the results and concepts would be overlapped with previous researches, the novelty given here would be enough. Here, I raise some issues to be answered below.

1. I understood that the authors extensively identified sweep region in Or vs Oc, with diversified peach accessions, and found good candidates. However, I was quite confused about how they could confine the sweep into "gene scale" regions. In peach, as suggested in many reports, LD do not decay in wide range, especially in modern cultivars (such as Or and Oc). For this situation, theoretically, haplotypes

cannot be separated in a large block. Did the authors imply that very recent minor independent SNPs (corresponding to specific core SNPs in XP-EHH) were directly located on the candidate genes? Otherwise, SFS-based methods (like π or CLR) showed specific reductions in the genic regions, in transition of the indexes in the objected sweep regions? In any cases, please make the logics clearer, and add appropriate figure(s).

2. Additionally, the authors lastly focused on expression correlation of PpERDL16 and fructose content, as a potential validation of the gene function under artificial selection. That means cis-motifs in the ERDL16 was a key to be under selection? If so, please show some clues for that. Of course, it may be difficult to identify motif-scale sequences under selections, while the authors should consider not only genic regions but also on cis-regulatory regions throughout the manuscript.

3. Partially relating to the last two queries, at glance, the selected genes in the selective sweep regions in Or vs Oc would look quite arbitrary. For instance, DAM gene showed no clear peak in both SFS- and EHH-based methods, while the highest peak such as in chr 2 were not focused at all.

4. I am quite sure that PpERDL16 expression is correlated to fructose content, while Figure 5b may look like that only three points with the relative expression level of PpERDL16 > 40 induced this strong correlation. Did these three act as outliers?

Minor points

1. L120: Rosaceae is a family name. Not Italics.
2. The second and third panels of Figure 3a were identical?
3. For Figure 3a, please provide criteria for definition of "ecological adaptation".

Reviewer #3 (Remarks to the Author):

In this manuscript, Yu et al. perform chromosome-scale genome assembly of a Chinese peach landrace. They then work on to identify candidate genes for key agronomic traits with genome-wide association using 564 accessions. This is in principle a very interesting experiment and may have significant application in breeding and agricultural practices.

The focus here on an improved assembly of the peach genome and genome-wide association analyses of several key agronomic traits is original and the analyses are generally competent. Here below, I will comment on some specific issues that particularly hampered my understanding.

1. In Section "A new high-quality LHSM reference genome", the authors have compared the assembled LHSM genome to the Lovell V2.0 genome. I am wondering what percentage of variants are actually due to assembly errors, especially errors in the Lovell genome assembly? For example, those structure variants in Fig.1c, could the authors show the alignment of Lovell Hi-C reads to both genomes as well?

2. The threshold value for the genome-wide significance in these association analyses has been determined with Bonferroni correction. This would not take into account of the distribution of the phenotypic data. I suggest the threshold values be determined based on permutation tests.

3. In the Supplementary Fig.10, the association peaks for the same phenotype collected in two different years overlapped badly, especially for Fructose content and Glucose content. Is there any particular reason for this?

Reviewer #4 (Remarks to the Author):

Modern peach breeding started in USA in the mid XIX century from a bunch of cultivars. One of them was the outstanding Chinese peach named 'Chinese Cling'. The modern western peach germplasm has this variety as main founder. The actual reference peach genome was established from a dihaploid completely homozygous accession from the rootstock 'Lovell'. The choosing of a rootstock as reference by the International Peach Genome Initiative was due mainly to the double haploid nature of the Lovell accession that assured a high-quality standard of the reference genome obtained. Nowadays the availability of long-reads sequencing (PacBio SMRT, Nanopore sequencing etc) together with innovative mapping approaches (optical mapping and Hi-C) allowed to obtain high quality sequences starting from heterozygous individuals. In this manuscript the author obtained a high standard sequence starting from a heterozygous accession of the Chinese Cling peach group. The sequence displays high quality standard in terms of completeness, base accuracy, and contiguity.

Moreover, a massive resequencing approach (564 accessions) using data obtained both by the authors and available in gene bank allowed to shed light on the path of peach domestication and breeding highlighting population structure and the selection progress for flavor traits (sweetness and acidity) in peach, from ancient to modern accessions. The association analysis pointed out the peculiarity of selection in the East (low acidity) and West (high acidity) identifying key genes underlying these important traits for fruit quality. The availability of a high quality LHSM genome allowed the comparison with the Lovell reference one, enabling the identification of large set of variants such as insertions, deletions, inversions, translocations and PAV. This is the first step toward the identification of a peach and *Prunus* pan and dispensable genomes. The authors also analyzed the evolution among 12 dicots plants including 7 Rosaceae species. This new genome sequence would serve the worldwide *Prunus* and rosaceae community as a valuable resource for advancement of genome analyses in this group of species.

I believe the manuscript deserves the publication in Nature Communications with the essential revisions reported below.

- In table 1 the comparison between LHSM and Lovell genomes is not homogenous for some genomic features.

The total n. of contigs in LHSM is reported as 243 but this is, as mentioned in Method "Genome Assembly" (pag 17), the total number of supercontigs. The corresponding number of supercontigs in Lovell genome, prior pseudomolecules build, is 241 (Verde et al 2017 supplementary table 12). The number of Contigs in LHSM genome is 2212, as reported in Method "Genome Assembly" pag 17. So, it would be better to have both features reported in the table: n of contigs (LHSM 2212 vs Lovell 2525) and n. of supercontigs (LHSM 243 vs Lovell 241).

Moreover, for the largest contig the table again compares inhomogeneous feature. The LHSM largest contig (18.8 Mb) is likely the largest supercontigs. The largest supercontigs in Lovell is 28.8 Mb mapped in pseudomolecule 1 (Verde et al 2017 Supplementary table 10).

The same happens for Contig N50 feature. LHSM Contig N 50, as reported in Methods Genome Assembly (pag 17) and in Supplementary Fig 1, is 686.03 kb compared with 255.5 KB of Lovell. I would also add the Supercontigs N50 to the table: LHSM 5.17 Mb vs 7.33 Mb in Lovell (Verde et al Supplementary table 12).

At the same time, I would change the text and the figure accordingly.

For the text at page 5 lines 92-93 I would change the 3 words "contigs" in "supercontigs2: "the assembly comprised 243 supercontigs with a supercontig N50 of 5.17 Mb. A total of 145 supercontigs,

which accounted for 95.7% (~246.0 Mb).

In Supplementary Fig 1 in the flowchart I would change "Contig N50: 5.17 Mb" in "Supercontig N50: 5.17 Mb".

- In Methods page 21 line 547 peach generation time is assumed of 7 years. I do not agree with this assumption. The paper cited (Xie et al 2017) do not report 7 years generation time. They report that the peach generation time is "not less than 3 years". This is correct but 7 years, in my opinion, is not acceptable. Generation time is defined as the average age when female give birth. 3-4 years should be the best estimate for peach. Please, reconsider your analysis and results using 3-4 generation time.

- At pag 11 line 275 there is a typo: "favor" should be "flavor".

- Pag 6-7 lines 144-148. In these lines the authors, comparing the LHSM and Lovell genomes, report a highly structured rearranged region at 13.31-18.86 Mb. I would be cautious about this statement opting for a more conservative and simple option. Rather than a genome rearrangement this could be related to mis-ordered or wrongly oriented scaffolds in Ch3 region in Lovell genome. In fact, in the peach v2 paper (Verde et al 2017) the authors report a large centromeric region on chromosome 3 (12-17.6 Mb) with a high suppression of recombination frequencies in all the linkage maps used for pseudomolecules generation. In this region six scaffolds spanning 7 Mb were placed in just 6 cM. As the authors stated some scaffolds in this region were ordered randomly because of the lack of recombination and even those ones ordered and oriented were placed with low probability due to the extensive suppression of recombination observed. The analysis of linkage maps obtained by different authors after the build of the Peach v2.0 assembly but prior the publication of Verde et al results pointed out putative mis-orders of scaffolds in the region, misorder that needed to be addressed in a future release, as stated in Peach v2 paper. I would avoid to state that large genome rearrangements occurred in that region. I rather pointed out putative mis-ordering or mis-orientation of scaffolds in that region in Lovell genome due to the extensive suppression of recombination as the peach v2 authors already stated. This aspect pointed out the importance of the long read sequencing and Hi-C approach to solve inconsistencies in regions with high suppression of recombination frequencies as the centromeric ones. According to this I would change the legend of Fig 1, removing the part related to the rearrangements. Moreover, using the LHSM genome and the extensive information in supplementary materials available in Verde et al 2017 the authors can address the putative mis-order and mis-orientation of the scaffolds in this region of Lovell genome assembly. I would also suggest reporting in a SI table the correct putative order and orientation of the scaffolds in Lovell genome in this Ch3 region.

- Gene annotation. Page 5 line 106. The number of protein-coding genes in LHSM genome is quite larger than that reported in Lovell genomes (35,215 vs 26,873, about 30% more). Apart from the different methodologies used the loci in Lovell genome were predicted when CDS overlapping with repeats is less than 20%. From the current manuscript I don't see any selection against overlapping repeats. This selection wipe out about 2000 gene models as reported at the PAG presentation by the authors (<https://pag.confex.com/pag/xxiii/webprogram/Paper14519.html>). In fact, if we compare the amount of gene models obtained in Lovell with the annotation method described in the LHSM manuscript and the number of gene models without repeat selections using the Verde et al annotation approach (26873 protein-coding genes + 2000 TE selected loci = ~29000 protein-coding genes) the number of Lovell gene models predicted in the current manuscript is similar to that obtained without the overlapping TE selection (~ 10% difference) by Verde et al. I would report that the large difference in the number of protein-coding genes among the two genomes assemblies is likely due a conservative selection against TE in Lovell genome.

- In Fig 1 E there is a typo: "Insertions" not "Insertoins"

Response to Reviewers' comments

Reviewer #1 (Remarks to the Author):

The authors describe a high quality peach reference genome, and the alignment of Illumina sequence to the reference. They identify genomic regions and genes correlated with sugar and acid content. They further showed the function of two proteins in sugar or acid accumulation in peach. Overall, the manuscript is well written and the results interesting to a wide audience. A few minor criticisms are listed below.

1. Define enzyme names in the abstract and line 82.

Response:

Thanks for the supportive comments about our manuscript. Regarding this comment specifically, we have defined enzyme names in the revised abstract and in line 82 (now Line 82–84).

2. Lines 36-37: “Fruit flavor quality is largely determined by the sweetness and acidity which are the most important factors influencing consumer preference and acceptability.” Although sweetness and acidity are important, flavor is much more complex than just sugars and acids.

Response:

Thanks for your reminding of our inappropriate description here; we have changed the expression in the revised manuscript (now Line 36–37), as:

“Sweetness and acidity are two of the important flavor determinants which influence consumer preference and acceptability.”

3. At least in the USA, of oriental and occidental are no longer used as descriptors of the East and West.

Response:

Thanks for this guidance. We have changed the description (from oriental and occidental to eastern and western, respectively) throughout the revised manuscript.

4. Sorbitol is found in a limited number of related stone fruits and is a significant contributor to sweetness in peaches, yet it is not focused on in the paper. GWAS analysis of sorbitol may lead to a better understanding of this sugar's contribution to flavor.

Response:

Thanks for your suggestion. We have now added information about our GWAS analysis for sorbitol content (data from 2016 and 2017). We provide PVEs (phenotypic variance explanations) for the significant signals in Supplementary Fig. 10. The added content in main text of the revised manuscript (Line 302–308) reads as follows:

“Sorbitol is universally found in stone fruits and is a significant contributor to sweetness in peaches^{19,70} (Walker *et al.*, 2020; Moing *et al.*, 1998). We detected two significant signals on chromosome 6 associated with sorbitol content in 2016, and one signal each for chromosome 1 and chromosome 3 in 2017; these respectively explained 7.5%, 8.1%, 7.9%, and 7.7% of phenotypic variation (Supplementary Fig. 11), thus identifying candidate loci for investigations about the genetic determinants of sorbitol accumulation. Of these, one signal on chromosome 6 (Chr6: 22,350,242 – 22,451,210) was found to overlap with a recently reported QTL for sorbitol⁷¹ (Cao *et al.*, 2019) (Supplementary Table 40).”

References:

1. Walker, R. P. *et al.* Non-structural carbohydrate metabolism in the flesh of stone fruits of the genus *Prunus* (Rosaceae)—A review. *Front. Plant Sci.* **11**, 549921 (2020).
2. Cao, K. *et al.* Comparative population genomics identified genomic regions and candidate genes associated with fruit domestication traits in peach. *Plant Biotechnol. J.* **17**, 1954–1970 (2019).

5. More references are needed in the methods section, for example Interpro, KEGG, BlastP, and other databases.

Response:

Thanks for your comment. We have added the citations (references) or URLs (now line 459) for these databases and software according to your guidance in the revised manuscript.

1. Blastp:

Altschul S. F., Gish W., Miller W., Myers E. W., & Lipman D. J. Basic local alignment search tool. *J Mol Biol.* **215**, 403–410 (1990).

2. Interpro:

Blum, M. et al. The InterPro protein families and domains database: 20 years on. *Nucleic Acids Res.* **49**, D344–D354 (2021).

3. KEGG:

Kanehisa, M., & Goto, S. KEGG: kyoto encyclopedia of genes and genomes. *Nucleic Acids Res.* **28**, 27–30 (2000).

4. RefSeq NR:

O'Leary, N. A. et al. Reference sequence (RefSeq) database at NCBI: current status, taxonomic expansion, and functional annotation. *Nucleic Acids Res.* **44**, D733–D745 (2016).

5. Pfam:

El-Gebali, S. et al. The Pfam protein families database in 2019. *Nucleic Acids Res.* **47**, D427–D432 (2019).

6. RepeatModeler:

(URL: <http://www.repeatmasker.org>)

7. RepeatMasker:

(URL: <http://www.repeatmasker.org>)

6. Line 620: I think a reference is added to the text here.

Response:

Yes, thanks for spotting this. We have removed it in the revised manuscript. Now the Line 628.

7. Line 626: remove highest

Response:

We have removed it in the revised manuscript. Now the Line 633.

8. Line 636: how many biological replicates?

Response:

We used three or more biological replicates in all of the PCR reactions. We now provide this information in our revised manuscript. Now the Line 643–644.

9. Line 642: Is it Redhaven?

Response:

Yes, thanks for spotting this. We have corrected it in the revised manuscript. Now the Line 650.

10. Table 1. Should it say Average transcript length.

Response:

Yes, thanks for your correction. We have changed the expression in Table 1 in our revised manuscript.

Thanks again for the helpful guidance to improve our study!

Reviewer #2 (Remarks to the Author):

Yu et al. drafted a new genome sequences with a landrace which is nested to representative commercial accessions (like Chinese Cling peach), and then utilized that for extensive population genetics with resequencing data from >500 accessions. They properly conducted selective sweep analyses with site frequency spectrum (SFS) and extended haplotype homozygosity (EHH)-based methods to identify the loci putatively under recent artificial selections in differentiation to geographical distribution. Furthermore, they applied extensive GWAS mainly to identify fruit quality-related loci, focusing acidity and sugar content. Lastly, they could identify an important gene undergoing artificial selection, named PpERDL16, of which expression level negatively contributes to fructose content in fruit. Overall, massive data was provided, including new draft genome sequences and >500 resequencing data. They looked properly interpreted by multiple aspects of population genetic approaches. Methods sound well. I highly appreciate the whole-genome sequencing of Chinese Cling-like accession, because the current standard, cv. Lovell peach, would be quite far from the modern cultivated peaches. Although some of the results and concepts would be overlapped with previous researches, the novelty given here would be enough. Here, I raise some issues to be answered below.

1. I understood that the authors extensively identified sweep region in Or vs Oc, with diversified peach accessions, and found good candidates. However, I was quite confused about how they could confine the sweep into “gene scale” regions. In peach, as suggested in many reports, LD do not decay in wide range, especially in modern cultivars (such as Or and Oc). For this situation, theoretically, haplotypes cannot be separated in a large block. Did the authors imply that very recent minor independent SNPs (corresponding to specific core SNPs in XP-EHH) were directly located on the candidate genes? Otherwise, SFS-based methods (like π or CLR) showed specific reductions in the genic regions, in transition of the indexes in the objected sweep regions? In any cases, please make the logics clearer, and add appropriate figure(s).

Response:

Thanks for your encouraging comments and helpful guidance. Regarding this comment specifically, window-scale sweep regions were obtained based on our analysis (see method); however, we agree with your comment that it is difficult to confine the sweep into “gene scale”

regions directly, especially when using an LD-based method. We understand that specific core variation(s) (e.g., SNP) can easily hijack nearby variants (Vitti et al., 2013) due to the linkage disequilibrium. And it is true that compared to some other self-incompatible fruit tree crops such as apple and pear, peach (self-compatibility) has relatively slow LD decay (See the figure below).

Linkage disequilibrium patterns for apple (a), pear (b), and peach (c)

We also understand your point about SFS-based methods, for which specific reductions have been shown in genic regions. Again, it is difficult to confine a “gene-scale” selection under this circumstance. Accordingly, seeking to be more accurate, we have now changed our formerly inappropriate descriptions referring to the putative selected genes within these sweep regions in the revised manuscript (Now Line 214–222):

“Within the selective sweep regions, we found genes plausibly related to the ecological adaptation of ECs and WCs to their specific habitats. Specifically, we found *Pp.LH.01G06551* (*DAM6*), a dormancy-associated MADS-box (*DAM*) gene^{50,51} (which functions in determining peach bud endodormancy^{52,53}) and genes putatively encoding VERNALIZATION1 (*VRN1*)⁵⁴, PSEUDO-RESPONSE REGULATOR 5 (*PRR5*)⁵⁵, FRIGIDA-like protein⁵⁶, and GLYCINE-RICH RNA-BINDING PROTEIN 7 (*GRP7*)⁵⁷ (Supplementary Table 32 and 33)—all of which have functions related to the regulation of flowering time in plants. This selection analysis highlights that genetic divergence may have shaped the habitat-specific adaptation of reproduction traits among the ECs and WCs.”

Note: We have changed the description for peach cultivars (from oriental cultivars (OrCs) and occidental cultivars (OcCs) to eastern cultivars (ECs) and western cultivars (WCs), respectively).

References:

Vitti, J. J., Grossman, S. R., & Sabeti, P. C. Detecting natural selection in genomic data. *Annu. Rev. Genet.* **47**, 97–120 (2013).

2. Additionally, the authors lastly focused on expression correlation of PpERDL16 and fructose content, as a potential validation of the gene function under artificial selection. That means cis-motifs in the ERDL16 was a key to be under selection? If so, please show some clues for that. Of course, it may be difficult to identify motif-scale sequences under selections, while the authors should consider not only genic regions but also on cis-regulatory regions throughout the manuscript.

Response:

Thanks for this comment. According to your helpful suggestion, we examined the $\theta\pi$ values of the upstream ~5 kb (regions potentially harbored cis-motifs) region of *PpERDL16*, and detected markedly reduced genetic diversity ($\theta\pi$ values) in this region (which also harbors the lead SNP (Chr1: 11,910,990) for the association of the fructose trait). Thus, our new analysis reveals a clue about its selection. We have now modified the figure (Supplementary Fig. 14) and added content to report this finding as follows (Now Line 348–351 in the revised manuscript):

“Intriguingly, we found that the $\theta\pi$ values of *PpERDL16* (both its CDS and the upstream (~ 5 kb) region harboring potential cis-regulatory elements) were lower among the modern cultivars (ECs or WCs) compared to PLs (Supplementary Fig. 14), again supporting improvement-related selection of *PpERDL16*.”

Supplementary Fig. 14 Reduced nucleotide diversity ($\theta\pi$) of *PpERDL16* during peach improvement. Displayed are the $\theta\pi$ values of consecutive windows (upper panel) and sites (lower panel) overlapping *PpERDL16*.

We then examined the significantly associated SNPs and InDels that may disturb the cis-regulatory elements within the upstream region ~5 kb region and found that there are in total 2 significant SNPs (including the lead SNP) and one significantly associated InDel located in the upstream region (~5 kb) of *PpERDL16*. However, none of these significant variations disturb the potential cis-regulatory elements (located in the cis-regulatory elements predicted using PlantCARE (<http://bioinformatics.psb.ugent.be/webtools/plantcare/html/>) (See the figure below). Further detection for SVs showed that no SVs share overlap with this region.

Figure Significantly associated SNPs and InDel and their nearby cis-regulatory elements within the upstream region (~5 kb) of *PpERDL16*. Potential cis-regulatory elements were marked as colored boxes.

3. Partially relating to the last two queries, at glance, the selected genes in the selective sweep regions in Or vs Oc would look quite arbitrary. For instance, DAM gene showed no clear peak in both SFS- and EHH-based methods, while the highest peak such as in chr 2 were not focused at all.

Response:

The key point for addressing this comment is the fact that our overall study (and indeed our research program generally) is focused on economic and horticultural traits in peach. Accordingly, we focused our research resources on searching for the candidate genes that may bring functional impacts on fruit flavor. Additionally, as the comparison were also carried out between OrCs (now ECs) and OcCs (now WCs), two different groups of peach accessions that are geographically isolated, we were also particularly interested in examining some candidate genes related to adaption traits, such as dormancy and flowering time related genes. Nevertheless, it is clear that gene(s) under strong selection signals, such as the peak in Chr2 you mentioned, may bring potential functional impacts on other agricultural traits. Accordingly, we have included all of the putative selected genes within the selective sweep regions in Supplementary Table 32 and 33, so that the wider research community can leverage this information for follow-on related studies.

4. I am quite sure that *PpERDL16* expression is correlated to fructose content, while Figure 5b may look like that only three points with the relative expression level of *PpERDL16* > 40 induced this strong correlation. Did these three act as outliers?

Response:

Thanks for the comment and helpful guidance. Exploring the idea of outliers, when we removed the data of the three accessions, the correlation between the *PpERDL16* expression and fructose content for the 34 accessions also showed significantly negative and comparable correlation coefficient (R); R = -0.51, $P = 1.9e-03$ (n = 34) compared to the former one result with 37 accessions (R = -0.56, $P = 3e-04$ (n = 37)) (see figure below).

A significantly negative correlation between fructose content and the *PpERDL16* expression level in 34 peach accessions and 37 peach accessions, respectively. The horizontal bars on the points represent the standard deviations.

Minor points

1. L120: Rosaceae is a family name. Not Italics.

Response:

Thanks for this correction. We have fixed this problem throughout the manuscript. Now the Line 41, 126, and 487, respectively.

2. The second and third panels of Figure 3a were identical?

Response:

Thanks for inquiring about this. The second and third panels of Figure 3a indeed are identical. In this study, we used multiple selective sweep identification methods for the comparisons of OrCs (query) vs. OcCs (control) and OcCs (query) vs. OrCs (control). Among these methods, the XP-EHH score from XP-EHH analysis is directional, but in many cases, don't really know which is the ancestral allele or not; thus, candidate windows figured by selscan were unidirectional. So the outputs obtained from both the comparisons of OrCs (query) vs. OcCs (control) and OcCs (query) vs. OrCs (control) are identical. We thus modified the Figure 3a by showing this panel for just one time.

3. For Figure 3a, please provide criteria for definition of “ecological adaptation”.

Response:

Thank for your helpful suggestion. We have provided the criteria of “ecological adaptation” in the Figure legend as:

“Selection signatures in the comparisons between ECs and WCs focused on ecological adaptation (specifically the regulation of dormancy and flowering time; blue labels) and acidity-related (red labels) candidate genes within their putative selective regions.”

We would like to take this opportunity to thank the reviewer for the helpful guidance about how to improve our study.

Reviewer #3 (Remarks to the Author):

In this manuscript, Yu et al. perform chromosome-scale genome assembly of a Chinese peach landrace. They then work on to identify candidate genes for key agronomic traits with genome-wide association using 564 accessions. This is in principle a very interesting experiment and may have significant application in breeding and agricultural practices.

The focus here on an improved assembly of the peach genome and genome-wide association analyses of several key agronomic traits is original and the analyses are generally competent. Here below, I will comment on some specific issues that particularly hampered my understanding.

1. In Section “A new high-quality LHSM reference genome”, the authors have compared the assembled LHSM genome to the Lovell V2.0 genome. I am wondering what percentage of variants are actually due to assembly errors, especially errors in the Lovell genome assembly? For example, those structure variants in Fig.1c, could the authors show the alignment of Lovell Hi-C reads to both genomes as well?

Response:

Thanks for focusing our attention here. We aligned the genome sequence of the LHSM genome onto Lovell v2.0 genome using MUMmer (Marcais et al., 2018), and the result showed that 89.62% of Lovell v2.0 genome could be aligned against the LHSM genome with a minimum identity of 90% (See table below). Among the aligned regions, up to 96.76% of sequences were found to have high synteny with the LHSM genome estimated using SyRI (Goel et al., 2019), supporting the high assembly quality of the Lovell v2.0 genome. The left aligned region (3.24%) had lower synteny, which may relate to rearrangement or assembly errors; we anticipate that future use of Hi-C data for both materials should help disambiguate this region. For example, as the reviewer mentioned, the large structural variants in **Fig.1c** could be partially validated using our LHSM Hi-C genome, markedly reducing the probability of an assembly error with the LHSM genome. However, confidently determining the source of these structure variants (*e.g.*, potential assembly errors in the Lovell v2.0 genome) will require more evidence like Hi-C data for the Lovell v2.0 genome. At present, we cannot assess potential assembly errors in the Lovell genome assembly as the doubled haploid ‘PLOV2-2N’ material (used for Lovell genome assembling) of the cultivar ‘Lovell’ is not

available to us.

The summary statistics for alignment of the LHSM genome against Lovell v2.0 genome

	LHSM	Lovell v2.0
Genome size	257,213,401 bp	227,411,381 bp
Size/percentage of aligned regions	239,390,236 bp/93.07%	203,805,040 bp/89.62%
Size/percentage of non-aligned regions	17,823,165 bp/6.93%	23,606,341 bp/10.38%
Size/percentage of syntenic regions	198,491,756 bp/82.92%	197,192,053 bp/96.76%

References:

1. Marçais, G. et al. MUMmer4: A fast and versatile genome alignment system. *PLoS Comput. Biol.* **14**, e1005944 (2018).
2. Goel, M., Sun, H., Jiao, W. B., & Schneeberger, K. SyRI: finding genomic rearrangements and local sequence differences from whole-genome assemblies. *Genome Biol.* **20**, 1–13 (2019).

2. The threshold value for the genome-wide significance in these association analyses has been determined with Bonferroni correction. This would not take into account of the distribution of the phenotypic data. I suggest the threshold values be determined based on permutation tests.

Response:

Thanks for this helpful comment. According to your suggestion, we performed the GWAS based on the permuted phenotypes. The distribution of the most significant *P* values across the 1000 replicates was used to determine the threshold; the range of the threshold values for all the traits we tested was from 6.24 to 6.97 (provided in the added Supplementary Table 37). We finally presented the thresholds from both these two tests (Permutation tests and Bonferroni correction) for all the traits we tested (see modified Figures in the revised manuscript). Seeking to further decrease the false positive rate, we chose the more stringent threshold from between these two tests for all of the traits that we studied.

Supplementary Table 37 The threshold of GWAS based on permutation test

Trait	Year	Threshold
Citrate	2016	6.68
Citrate	2017	6.59
TA	2017	6.64
Malate	2016	6.97
Malate	2017	6.60
pH	2016	6.31
pH	2017	6.26
Fructose	2016	6.35
Fructose	2017	6.24
Glucose	2016	6.63
Glucose	2017	6.42
Sorbitol	2016	6.53
Sorbitol	2017	6.55
Sucrose	2016	6.39
Sucrose	2017	6.51

3. In the Supplementary Fig. 10, the association peaks for the same phenotype collected in two different years overlapped badly, especially for Fructose content and Glucose content. Is there any particular reason for this?

Response:

Thanks for this comment. Generally speaking, some of the association signals do not overlap well between the two years, reflecting their nature as environmentally sensitive quantitative traits. And notably, previous studies have used year labels for some peach fruit quality-related QTLs (<https://www.rosaceae.org/search/qtl>).

Of particular note with peach, it has been recorded that except for sucrose content—which seems to be relatively less influenced by environmental variables—variation in glucose, fructose, and sorbitol content can display significant changes between years (Brooks et al. 1993). The variation in sugar content between years can likely be explained by differences in environmental conditions (Culpepper and Caldwell, 1930). Despite this, year-stable signal(s) have been reported for sugar related traits in peach, e.g. sorbitol content (Cao et al., 2019). Kindly note that in the present study, we primarily focused our attention on year-stable loci (peaks) for mining of causal

genes, *e.g.*, *PpERDL16*; yet we have also provided the information for other loci which may contribute to particular traits as a reference for relevant studies in the future.

References:

1. Brooks, S.J., Moore, J.N., & Murphy, J.B. Quantitative and qualitative changes in sugar content of peach genotypes [*Prunus persica* (L.) Batsch.]. *J. AM. SOC. HORTIC. SCI.* **1**, 97–100 (1993).
2. Culpepper, C.W., & Caldwell, J.S. The canning quality of certain commercially important eastern peaches. United States Department of Agricultural Technology. Bul. 196 (1930).
3. Cao, K. et al. Comparative population genomics identified genomic regions and candidate genes associated with fruit domestication traits in peach. *Plant Biotechnol. J.* **17**, 1954–1970 (2019).

We appreciate the helpful guidance about how to make our paper better. Many thanks.

Reviewer #4 (Remarks to the Author):

Modern peach breeding started in USA in the mid XIX century from a bunch of cultivars. One of them was the outstanding Chinese peach named 'Chinese Cling'. The modern western peach germplasm has this variety as main founder. The actual reference peach genome was established from a dihaploid completely homozygous accession from the rootstock 'Lovell'. The choosing of a rootstock as reference by the International Peach Genome Initiative was due mainly to the double haploid nature of the Lovell accession that assured a high-quality standard of the reference genome obtained. Nowadays the availability of long-reads sequencing (PacBio SMRT, Nanopore sequencing etc) together with innovative mapping approaches (optical mapping and Hi-C) allowed to obtain high quality sequences starting from heterozygous individuals. In this manuscript the author obtained a high standard sequence starting from a heterozygous accession of the Chinese Cling peach group. The sequence displays high quality standard in terms of completeness, base accuracy, and contiguity.

Moreover, a massive resequencing approach (564 accessions) using data obtained both by the authors and available in gene bank allowed to shed light on the path of peach domestication and breeding highlighting population structure and the selection progress for flavor traits (sweetness and acidity) in peach, from ancient to modern accessions. The association analysis pointed out the peculiarity of selection in the East (low acidity) and West (high acidity) identifying key genes underlying these important traits for fruit quality. The availability of a high quality LHSM genome allowed the comparison with the Lovell reference one, enabling the identification of large set of variants such as insertions, deletions, inversions, translocations and PAV. This is the first step toward the identification of a peach and *Prunus* pan and dispensable genomes. The authors also analyzed the evolution among 12 dicots plants including 7 Rosaceae species. This new genome sequence would serve the worldwide *Prunus* and rosaceae community as a valuable resource for advancement of genome analyses in this group of species.

I believe the manuscript deserves the publication in Nature Communications with the essential revisions reported below.

- In table 1 the comparison between LHSM and Lovell genomes is not homogenous for some genomic features.

The total n. of contigs in LHSM is reported as 243 but this is, as mentioned in Method “Genome Assembly” (pag 17), the total number of supercontigs. The corresponding number of supercontigs in Lovell genome, prior pseudomolecules build, is 241 (Verde et al 2017 supplementary table 12). The number of Contigs in LHSM genome is 2212, as reported in Method "Genome Assembly" pag 17. So, it would be better to have both features reported in the table: n of contigs (LHSM 2212 vs Lovell 2525) and n. of supercontigs (LHSM 243 vs Lovell 241).

Moreover, for the largest contig the table again compares inhomogeneous feature. The LHSM largest contig (18.8 Mb) is likely the largest supercontigs. The largest supercontigs in Lovell is 28.8 Mb mapped in pseudomolecule 1 (Verde et al 2017 Supplementary table 10).

The same happens for Contig N50 feature. LHSM Contig N 50, as reported in Methods Genome Assembly (pag 17) and in Supplementary Fig 1, is 686.03 kb compared with 255.5 KB of Lovell. I would also add the Supercontigs N50 to the table: LHSM 5.17 Mb vs 7.33 Mb in Lovell (Verde et al Supplementary table 12).

At the same time, I would change the text and the figure accordingly.

For the text at page 5 lines 92-93 I would change the 3 words “contigs” in “supercontigs2: *“the assembly comprised 243 supercontigs with a supercontig N50 of 5.17 Mb. A total of 145 supercontigs, which accounted for 95.7% (~246.0 Mb).”*

In Supplementary Fig 1 in the flowchart I would change “*Contig N50: 5.17 mB*” in “*Supercontig N50: 5.17 Mb*”.

Response:

Thanks for the positive and encouraging comments about our study. Following your helpful guidance, we have now carefully investigated the data in the paper that updated the Lovell reference genome (Verde et al., 2017). In that paper, the author mentioned that for the Lovell v2.0 assembly the scaffold N50 is 7.3 Mb and the contig N50 is 255.4 kb. However, in the LHSM assembly the “supercontig” is actually different from the “scaffold” (which was composed of

contigs and gaps based on a genetic map) (see table below); the “supercontig” is a contig that was extended using corrected PacBio long reads with the HERA algorithm (and does not contain gaps). HERA can be used to connect and fill gaps between CANU-assembled contigs, specifically by traversing the overlap graph with corrected reads. As such, at least according to previous studies (Shen *et al.*, 2019; Shen *et al.*, 2020) and our understanding, it is suitable to refer to products from HERA as “contigs”, and these contigs have no gaps. To avoid any misunderstanding, we have changed “supercontig” to “contigs” throughout the revised manuscript (including the Method, Table 1, and Supplementary Fig.1).

Statistics for the LHSM genome assembly in comparison with the Lovell v2.0

Genomic feature	LHSM	Lovell v2.0
Sequenced genotype	Diploid	Double haploid
Total assembly size	257.2 Mb	227.4 Mb ^a
Number of contigs	243 ^b	2,525 ^a
Largest contig	18.8 Mb ^b	1.5 Mb ^a
Contig N50 length	5.17 Mb ^b	255.4 kb ^a
Largest scaffold	--	28.8 Mb
Scaffold N50	--	7.3 Mb
Sequences anchored to chromosomes	246.0 Mb	225.7 Mb ^a

^a The statistic values taken from the previous publication³².

^b Contigs assembled using HERA method.

References:

1. Shen Y. et al. Update soybean Zhonghuang 13 genome to a golden reference. *Sci China Life Sci.* **62**, 1257–1260 (2019).
2. Shen C. et al. The chromosome-level genome sequence of the autotetraploid Alfalfa and resequencing of core germplasms provide genomic resources for Alfalfa research. *Mol. Plant* **13**, 1250–1261 (2020).

• In Methods page 21 line 547 peach generation time is assumed of 7 years. I do not agree with this assumption. The paper cited (Xie et al 2017) do not report 7 years generation time. They report that the peach generation time is “not less than 3 years”. This is correct but 7 years, in my

opinion, is not acceptable. Generation time is defined as the average age when female give birth. 3-4 years should be the best estimate for peach. Please, reconsider your analysis and results using 3-4 generation time.

Response:

Thanks for your suggestion about this particular issue. We have now re-analyzed our data following the suggestion of using generation times of 3 and 4 years (See new Figure 2e and Supplementary Fig. 5). The trends detected in this analysis were similar to our original analysis.

Figure 2e. Demographic analysis for different groupings of peach accessions. A multiple sequentially Markovian coalescent (MSMC) model was used to infer their effective population fluctuations under a mutation rate $\mu = 7.7 \times 10^{-9}$ per site per generation (Xie *et al.*, 2016) and the generation time of 3 years (and generation time of 4 years in Supplementary Fig. 5).

- At pag 11 line 275 there is a typo: “favor” should be “flavor”.

Response:

Thanks for spotting this. We have corrected this typo in our revised manuscript (Now Line 280).

- Pag 6-7 lines 144-148. In these lines the authors, comparing the LHSM and Lovell genomes, report a highly structured rearranged region at 13.31-18.86 Mb. I would be cautious about this statement opting for a more conservative and simple option. Rather than a genome rearrangement this could be related to mis-ordered or wrongly oriented scaffolds in Ch3 region in Lovell genome. In fact, in the peach v2 paper (Verde et al 2017) the authors report a large centromeric region on

chromosome 3 (12–17.6 Mb) with a high suppression of recombination frequencies in all the linkage maps used for pseudomolecules generation. In this region six scaffolds spanning 7 Mb were placed in just 6 cM. As the authors stated some scaffolds in this region were ordered randomly because of the lack of recombination and even those ones ordered and oriented were placed with low probability due to the extensive suppression of recombination observed. The analysis of linkage maps obtained by different authors after the build of the Peach v2.0 assembly but prior the publication of Verde et al results pointed out putative mis-orders of scaffolds in the region, disorder that needed to be addressed in a future release, as stated in Peach v2 paper. I would avoid to state that large genome rearrangements occurred in that region. I rather pointed out putative mis-ordering or mis-orientation of scaffolds in that region in Lovell genome due to the extensive suppression of recombination as the peach v2 authors already stated. This aspect pointed out the importance of the long read sequencing and Hi-C approach to solve inconsistencies in regions with high suppression of recombination frequencies as the centromeric ones.

According to this I would change the legend of Fig 1, removing the part related to the rearrangements. Moreover, using the LHSM genome and the extensive information in supplementary materials available in Verde et al 2017 the authors can address the putative mis-order

and mis-orientation of the scaffolds in this region of Lovell genome assembly. I would also suggest reporting in a SI table the correct putative order and orientation of the scaffolds in Lovell genome in this Ch3 region.

Response:

Thanks for your helpful suggestions. We agree that it is more plausible that these putative mis-orderings or mis-orientations of scaffolds in the Lovell genome region in question could reflect extensive suppression of recombination as the previous study stated (Verde et al 2017). According to your guidance, we have now examined the corresponding region at 13.31–18.86 Mb using synteny analysis between the LHSM genome assembly and the scaffolds of the Lovell v2.0 genome, and provided this information as a supplementary table (Supplementary Table 21). We have modified the description in the revised manuscript (Now line 153–158) as follows:

“Thus, we further examined this region based on a comparison of the Hi-C contact matrices for the LHSM and Lovell v2.0 assemblies constructed using the LHSM Hi-C data (Fig. 1b,c and

Supplementary Table 18) as well as through synteny analysis between the LHSM genome assembly and the scaffolds of the Lovell v2.0 genome (Supplementary Table 21). The results suggest mis-ordering or mis-orientation of scaffolds in the corresponding region of the Lovell v2.0 genome.”

Supplementary Table 21 The synteny alignment between the structured rearranged region of LHSM and scaffold of the Lovell v2.0 genome

Code	LHSM		Lovell v2.0		
	Chromosome	Position on chromosome	Scaffold	Position on Scaffold	Putative mis-orientation or mis-ordering
I	Chr3	13,312,855–13,870,436	Super_32	575,509–45,224	mis-orientation
II	Chr3	14,334,748–15,228,781	Super_18	973,342–659	mis-orientation
III	Chr3	15,230,244–16,451,437	Super_18	2,092,361–1,008,570	mis-orientation
IV-1	Chr3	16,628,662–17,336,800	Super_27	122,428–786,082	mis-ordering
IV-2	Chr3	17,337,291–18,863,662	Super_451	3–1,445,406	mis-ordering

- Gene annotation. Page 5 line 106. The number of protein-coding genes in LHSM genome is quite larger than that reported in Lovell genomes (35,215 vs 26,873, about 30% more). Apart from the different methodologies used the loci in Lovell genome were predicted when CDS overlapping with repeats is less than 20%. From the current manuscript I don’t see any selection against overlapping repeats. This selection wipe out about 2000 gene models as reported at the PAG presentation by the authors (<https://pag.confex.com/pag/xxiii/webprogram/Paper14519.html>). In fact, if we compare the amount of gene models obtained in Lovell with the annotation method described in the LHSM manuscript and the number of gene models without repeat selections using the Verde et al annotation approach (26873 protein-coding genes +2000 TE selected loci = ~29000 protein-coding genes) the number of Lovell gene models predicted in the current manuscript is similar to that obtained without the overlapping TE selection (~ 10% difference) by Verde et al. I would report that the large difference in the number of protein-coding genes among the two genomes assemblies is likely due a conservative selection against TE in Lovell genome.

Response:

Thanks for this excellent guidance. We have now performed an analysis of overlapping TEs with CDS, and found 10,118 protein-coding genes, for which the percentage of CDS overlap with TEs

was 28.7% on average. We have added a supplementary table (Supplementary Table 10) and following text to the revised manuscript (Now Line 111–116):

“An analysis of TEs overlap with CDS regions indicated TEs overlap for 10,118 protein-coding genes; the percentage of CDS overlapped by TEs was 28.7% on average (Supplementary Table 10). Apart from the different methodologies used, the large difference in the number of protein-coding genes between the LHSM and Lovell v2.0 genome assemblies is likely due to a conservative selection criterion against TE in the Lovell genome: their pipeline used an overlap value of less than 20% for TEs overlap of CDS regions³¹ (Verde et al., 2013).”

Reference:

Verde, I. et al. The high-quality draft genome of peach (*Prunus persica*) identifies unique patterns of genetic diversity, domestication and genome evolution. *Nat Genet.* **45**, 487–494 (2013).

- In Fig 1 E there is a typo: “Insertions” not “Insertoins”

Response:

Thanks for spotting this. We have corrected this typo in Fig. 1e.

We would like to take this opportunity to thank the reviewer for the supportive comments and the excellent guidance about how to improve our study.

REVIEWERS' COMMENTS

Reviewer #1 (Remarks to the Author):

The authors have done an excellent job of responding to the reviewers' comments. I feel the manuscript is ready for publication.

Reviewer #2 (Remarks to the Author):

The authors have made some revisions/comments (and additional figures) against my queries, but they were not properly interpreted. Overall, my question was depended on the specific nature of the peach genome, such as very long LDs. As I expected, LDs in the peach genome were not decayed over 1000-kb as exhibited in the rebuttal letter (and this would not be due to self-compatibility, but to the recent limited founder effects). Thus, it is actually hard to finely confine the genes under artificial selection, although that would be the issue.

For the authors revisions, line 214-222, still selection of the genes in the main text would look arbitrary. For instance, "all of which have functions related to the regulation of flowering time in plants" would be ambiguous, and based on very limited selections from the sweep region.

Supplementary Fig. 14 would not suggest that the cis-regulatory region of the PpERDL16 is specifically under the selection. At least 50-kb regions in the Supplementary Fig. 14, there would be no substantial turning points for selective pressures (or historical recombination). As I suggested, in long-haploblocks formed recently, wide regions should show similar genetic diversities, which would be hard to utilize for confining the selected genome region finely. That would be no problem, but it would be unfavorable to make a (often arbitrary) conclusion from the data which is not appropriate to squeeze that.

Regarding the rebuttal comments, "The key point for addressing this comment is the fact that our overall study (and indeed our research program generally) is focused on economic and horticultural traits in peach. Accordingly, we focused our research resources on searching for the candidate genes that may bring functional impacts on fruit flavor", that would be OK, but this should not be the reason for that the authors can define the "artificially selected genes" from abundant candidates. The point in my previous comment was "the authors should not arbitrarily pick up a gene (DAM gene) with no clear selective peaks both in SFS- and EHH-based methods", but no response about that. Discussion based on your research project (here economic and horticultural traits) would be no problem, while definition of "selected genes" should be objectively based on the results. At least in the currently provided results, there would be no clear reasons to define the DAM genes as under selection.

Reviewer #3 (Remarks to the Author):

The authors have addressed all reviewer comments. I suggest the paper should get accepted.

Reviewer #4 (Remarks to the Author):

The authors answer to all my questions and follow most of my suggestions. I particularly appreciate the SI tables they included and the filter they adopted on their annotation pipeline for the TE overlapping genes.

I have a couple of more suggestions.

I understand that the definition of superconting (or scaffold), with the new assembly tools and methods, does not fit with the actual LHSM assembly so I agree that the best way is to call "contig" the results of the Hera assembly (ungapped). However, supercontigs (or scaffolds, they are synonymous) are not based on linkage map, pseudomolecules (or pseudochromosomes) are. Supercontigs, unless contigs, are, by definition, gapped and they are the results of the second round of assembly (scaffolding) based on paired end sequencing. Given that, I think the best way to compare the two assemblies, that were obtained with different approaches, is to give all the information available for the two assemblies. So, I suggest including in Table 1 the information that the authors report in the confidential table to reviewers adding two extra lines: Largest scaffold (LHSM --; Lovell 28.8 Mb) and Scaffold N50 (LHSM --; Lovell 7.3 Mb)

About the situation on chromosome 3 I appreciate the modifications of the text, new figure 3 and the inclusion of SI Table 21. However, I would credit in the text Verde et al 2017 to have suggested the region complexity in Lovell genome adding at line 158 of the revised MS something such as "as already highlighted by Verde et al 2017". Apart the indication of the complex and unclear situation in that region they indicated, for example, that Super_27 and Super_451 were misordered and their order in the pseudomolecule needed to be inverted in a future release, as reported in the text and in SI Table 10, indication confirmed with the LHSM assembly.

Response to Reviewers' comments

Reviewer #1 (Remarks to the Author):

The authors have done an excellent job of responding to the reviewers' comments. I feel the manuscript is ready for publication.

Response:

We thank the reviewer for the guidance and care in helping us to improve our manuscript.

Reviewer #2 (Remarks to the Author):

The authors have made some revisions/comments (and additional figures) against my queries, but they were not properly interpreted. Overall, my question was depended on the specific nature of the peach genome, such as very long LDs. As I expected, LDs in the peach genome were not decayed over 1000-kb as exhibited in the rebuttal letter (and this would not be due to self-compatibility, but to the recent limited founder effects). Thus, it is actually hard to finely confine the genes under artificial selection, although that would be the issue.

For the authors revisions, line 214-222, still selection of the genes in the main text would look arbitrary. For instance, “all of which have functions related to the regulation of flowering time in plants” would be ambiguous, and based on very limited selections from the sweep region.

Response:

Thanks for the comment. Considering the natural history of peach and the specific attributes of the peach genome—which brings difficulty for finely confining the genes under artificial selection—and seeking to avoid any arbitrary conclusions, we have now removed the descriptions (See line 217) (and corresponding information in Figure 3a) about the specific genes apparently localized in the selective sweep regions.

Supplementary Fig. 14 would not suggest that the cis-regulatory region of the PpERDL16 is specifically under the selection. At least 50-kb regions in the Supplementary Fig. 14, there would be no substantial turning points for selective pressures (or historical recombination). As I suggested, in long-haploblocks formed recently, wide regions should show similar genetic diversities, which would be hard to utilize for confining the selected genome region finely. That would be no problem, but it would be unfavorable to make a (often arbitrary) conclusion from the data which is not appropriate to squeeze that.

Response:

Thanks for this comment. It is difficult to confine the reduced diversity of the cis-regulatory region of *PpERDL16* as resulting from selection or lacking recent recombination. Accordingly, we have now toned down the statements in the revised manuscript, which now read as follows:

“We found that the $\theta\pi$ values of *PpERDL16* (both its CDS and the upstream (~ 5 kb) region harboring potential cis-regulatory elements) were lower among the modern cultivars (ECs or WCs) compared to PLs (Supplementary Fig. 18), which could hypothetically have resulted from selection for *PpERDL16*.”

Regarding the rebuttal comments, “The key point for addressing this comment is the fact that our overall study (and indeed our research program generally) is focused on economic and horticultural traits in peach. Accordingly, we focused our research resources on searching for the candidate genes that may bring functional impacts on fruit flavor”, that would be OK, but this should not be the reason for that the authors can define the “artificially selected genes” from abundant candidates. The point in my previous comment was “the authors should not arbitrarily pick up a gene (DAM gene) with no clear selective peaks both in SFS- and EHH-based methods”, but no response about that. Discussion based on your research project (here economic and horticultural traits) would be no problem, while definition of “selected genes” should be objectively based on the results. At least in the currently provided results, there would be no clear reasons to define the DAM genes as under selection.

Response:

Thanks for the comment. Again, to avoid any arbitrary conclusion, we have removed the descriptions (See line 217) (and corresponding information in Figure 3a) about the specific genes apparently localized in selective sweep regions. Furthermore, we changed the titles of Supplementary Data 17 and 18 from “Protein-coding genes within the selective sweep regions...” to “Possible genes under selection by comparing of ...” and emphasize the hypothetical nature of considering them as candidate “selected genes” in the revised manuscript and relevant figure legend (Figure 3).

Reviewer #3 (Remarks to the Author):

The authors have addressed all reviewer comments. I suggest the paper should get accepted.

Response:

We appreciate the reviewer’s support and are thankful for the guidance about our study.

Reviewer #4 (Remarks to the Author):

The authors answer to all my questions and follow most of my suggestions. I particularly appreciate the SI tables they included and the filter they adopted on their annotation pipeline for the TE overlapping genes.

I have a couple of more suggestions.

I understand that the definition of superconting (or scaffold), with the new assembly tools and methods, does not fit with the actual LHSM assembly so I agree that the best way is to call “contig” the results of the Hera assembly (ungapped). However, supercontigs (or scaffolds, they are synonymous) are not based on linkage map, pseudomolecules (or pseudochromosomes) are. Supercontigs, unless contigs, are, by definition, gapped and they are the results of the second round of assembly (scaffolding) based on paired end sequencing. Given that, I think the best way to compare the two assemblies, that were obtained with different approaches, is to give all the information available for the two assemblies. So, I suggest including in Table 1 the information that the authors report in the confidential table to reviewers adding two extra lines: Largest scaffold (LHSM --; Lovell 28.8 Mb) and Scaffold N50 (LHSM --; Lovell 7.3 Mb)

Response:

We appreciate your ongoing support, and are thankful for your continued guidance in improving our study. We have now complemented the comparison between the two assemblies by adding the terms of Largest scaffold (LHSM --; Lovell 28.8 Mb) and Scaffold N50 (LHSM --; Lovell 7.3 Mb) in Table 1.

Table 1 Summary statistics for the LHSM genome assembly in comparison with the Lovell v2.0 reference genome

Genomic feature	LHSM	Lovell v2.0
Sequenced genotype	Diploid	Double haploid
Total assembly size	257.2 Mb	227.4 Mb ^c
Number of contigs	243 ^d	2,525 ^c
Largest contig	18.8 Mb ^d	1.5 Mb ^c
Contig N50 length	5.17 Mb ^d	255.4 kb ^c
Largest scaffold	--	28.8 Mb ^c
Scaffold N50	--	7.3 Mb ^c
Sequences anchored to chromosomes	246.0 Mb	225.7 Mb ^c
GC content	37.57%	37.05%
Number of gaps ^a	137	1,828
Complete BUSCOs ^b	97.4%	96.8%
LTR assembly index, LAI	20.67	21.29
Repetitive sequences	118.35 Mb/46.01%	101.99 Mb/44.85%
Protein-coding genes/transcripts	35,215/40,072	31,972/47,089

Average transcript length

2,175 bp

2,215 bp

^a Gaps defined as >10 Ns.

^b The analysis from comparisons with the eudicotyledons_odb10 database.

^c The statistic values taken from the previous publication³².

^d Contigs assembled using HERA method.

About the situation on chromosome 3 I appreciate the modifications of the text, new figure 3 and the inclusion of SI Table 21. However, I would credit in the text Verde et al 2017 to have suggested the region complexity in Lovell genome adding at line 158 of the revised MS something such as “as already highlighted by Verde et al 2017”. Apart the indication of the complex and unclear situation in that region they indicated, for example, that Super_27 and Super_451 were misordered and their order in the pseudomolecule needed to be inverted in a future release, as reported in the text and in SI Table 10, indication confirmed with the LHSM assembly.

Response:

Thanks for this suggestion. We have now added the context about this issue in the revised manuscript as follows:

“Beyond showing the complexity of this region in the Lovell genome which —was highlighted by Verde *et al.* (2017)³²—these results supported the putative mis-ordering or mis-orientation of some scaffolds in the corresponding region of the Lovell v2.0 genome; for example, the Super_27 and Super_451 were misordered, and their order in the pseudomolecule should be inverted in a future release (Supplementary Table 16).”